# *PCDHA9* as a candidate gene for amyotrophic lateral sclerosis

Jie Zhong[1,2,10], Chaodong Wang [3,10] ✉, Dan Zhang [1,2,10], Xiaoli Yao [4,10], Quanzhen Zhao[5], Xusheng Huang[6], Feng Lin[7], Chun Xue [1,2], Yaqing Wang [1,2], Ruojie He[4], Xu-Ying Li[3], Qibin Li[8], Mingbang Wang [9], Shaoli Zhao[1,2], Shabbir Khan Afridi [1,2], Wenhao Zhou [9], Zhanjun Wang[3], Yanming Xu [5] ✉ & Zhiheng Xu [1,2] ✉

Amyotrophic lateral sclerosis (ALS) is a devastating neurodegenerative disease. To identify additional genetic factors, we analyzed exome sequences in a large cohort of Chinese ALS patients and found a homozygous variant (p.L700P) in *PCDHA9* in three unrelated patients. We generated *Pcdhα9* mutant mice harboring either orthologous point mutation or deletion mutation. These mice develop progressive spinal motor loss, muscle atrophy, and structural/functional abnormalities of the neuromuscular junction, leading to paralysis and early lethality. TDP-43 pathology is detected in the spinal motor neurons of aged mutant mice. Mechanistically, we demonstrate that *Pcdha9* mutation causes aberrant activation of FAK and PYK2 in aging spinal cord, and dramatically reduced NKA-α1 expression in motor neurons. Our single nucleus multi-omics analysis reveals disturbed signaling involved in cell adhesion, ion transport, synapse organization, and neuronal survival in aged mutant mice. Together, our results present *PCDHA9* as a potential ALS gene and provide insights into its pathogenesis.

Amyotrophic lateral sclerosis (ALS) is a fatal neurodegenerative disorder, with a prevalence of ~2/100,000 worldwide[1–3], and a lifetime risk of ~1 in 300 in Western populations[4,5]. Most ALS cases are sporadic, whereas 10% are familial[6]. Although there was no statistical data on the incidence of ALS in China, the annual incidence of ALS is 0.51/100,000/year in Taiwan[7] and 0.31–0.6/100,000/year in Hong Kong[8]. It is lower than that in Japan (2.2/100,000/year)[9] and Europe (2.16/100,000/year)[10]. Moreover, the percentage of fALS is only 1.2–2.7%[11] and the proportion of bulbar-onset patients is only 14% in Chinese patients[12]. ALS is characterized by the progressive degeneration of

motor neurons in the motor cortex, brain stem, and spinal cord, which gradually leads to paralysis and death from respiratory failure[5]. ALS patients usually die within 5 years from diagnosis[13–16].

Both genetic and environmental factors contribute to the etiology of ALS. So far, more than 120 genes have been reported with varying levels of evidence including many with weak/limited evidence associated with ALS (http://alsod.iop.kcl.ac.uk)[4]. About 30 genes were shown to have strong evidence associated with familial ALS (fALS), sporadic ALS (sALS), or both[5,17–19]. However, these genes account for only ~15% of sALS and ~80% of fALS[4,20–22]. Among them, *C9ORF72, SOD1,*

[1]State Key Laboratory of Molecular Developmental Biology, Institute of Genetics and Developmental Biology, Chinese Academy of Sciences, Beijing 100101, China. [2]University of Chinese Academy of Sciences, Beijing 100101, China. [3]Department of Neurology, Xuanwu Hospital, Capital Medical University, National Clinical Research Center for Geriatric Disease, Beijing 100053, China. [4]Department of Neurology, The First Affiliated Hospital, Sun Yat-sen University, Guangzhou 510080, China. [5]Department of Neurology, West China Hospital, Sichuan University, Chengdu 610041, China. [6]Department of Neurology, The First Medical Center, Chinese PLA General Hospital, Beijing 100853, China. [7]Department of Neurology, Fujian Medical University Union Hospital, Fuzhou 350001, China. [8]Shenzhen Clabee Biotechnology Incorporation, Shenzhen 518057, China. [9]Shanghai Key Laboratory of Birth Defects, Division of Neonatology, Children's Hospital of Fudan University, National Center for Children's Health, Shanghai 201102, China. [10]These authors contributed equally: Jie Zhong, Chaodong Wang, Dan Zhang, Xiaoli Yao. ✉e-mail: cdongwang@xwhosp.org; neuroxym@scu.edu.cn; zhxu@genetics.ac.cn

*TARDBP*, and *FUS* are recognized as the most common ALS genes. *C9ORF72* hexanucleotide repeat expansion (HRE), (GGGGCC)$_n$, the most common mutation in ALS[22–24], is rare in Eastern Asian ALS patients[25,26], suggesting genetic heterogeneity across ethnic populations and regions.

Mouse models have provided insights into the pathogenic molecular mechanisms of many diseases[22,27–29]. However, most ALS mouse models investigated so far are over-expression models of ALS-associated genes[30–33]. The extent of genetic alteration required to induce pathology in these studies makes their relationship to human disease pathogenesis uncertain[30]. Only a few knock-in models (expressing physiological levels of mutant genes) show ALS phenotypes. *Matr3* knock-in mice showed many typical ALS-like phenotypes[34] while *Dctn1* and *Vcp* knock-in mice mimic partial phenotypes[35,36]. Thus, the identification of genes contributing to ALS and the generation of robust mouse models that display the clinical and neurodegenerative features of ALS will help spur disease insights and therapies development.

*PCDHA9* (protocadherin alpha 9) is a member of the *PCDHα* family, which is one of the protocadherin gene clusters (*PCDHα, PCDHβ,* and *PCDHγ*) with their expression mainly restricted to the center nervous system[37–43]. The clustered protocadherin plays roles as neuronal barcodes to influence self-recognition and self-avoidance during the assembly of functional neural circuits. The clustered protocadherin represents the largest group within the cadherin superfamily[44–46]. Fourteen members of the PCDH-α family are expressed stochastically (α1-α12) or constitutively (αc1 and αc2) in single neurons in mice[47,48]. They have the same intracellular constant domain in the C-terminal and highly similar sequences (repeated extracellular cadherin [EC] motifs) in the N-terminal variable domains[40,49,50]. Nucleotide variants and aberrant alterations in DNA methylation in the *PCDH* gene clusters have been implicated in a variety of neurodevelopmental (Down syndrome) or psychiatric disorders (autism) and congenital heart disease (CHD)[15,41,49,51–53]. Nevertheless, there is little definitive evidence for a causal relationship between *PCDH-α* and diseases, although a *PCDHA9* variant was detected in human Hirschsprung's disease (Shen and Zhan, 2018) and a mutation in *Pcdhα9* was shown to contribute to CHD in mice[54].

In this study, we detected a recurrent homozygous variant in *PCDHA9* (c.2099 T > C; p.Leu700Pro) in a cohort of Chinese ALS cases. We generated both homozygous *Pcdhα9* missense mutation and frameshift deletion mice and found that *Pcdhα9* is essential for the survival of motor neurons and maintenance of the neuromuscular junction (NMJ). These mice exhibited ALS-like phenotypes, evidence that *PCDHA9* could contribute to ALS. We found that the ALS-associated L700P variant destabilizes the PCDHA9 protein and decreases the levels of its interacting protein Na$^+$/K$^+$ ATPase-α1. Multiomics analysis of the spinal cord of *Pcdha9* mutant animals reveals dysregulation of gene and protein expression levels associated with cell/cell-cell adhesion, ion transport, and neuronal survival. Together, the discovery of the *PCDHA9* variant in ALS and their validation in mouse models reveal further insight into the pathogenesis of ALS.

## Results

### Identification of a rare damaging homozygous PCDHA9 variant in Chinese sporadic ALS patients

To screen for mutations in ALS, we performed whole-exome sequencing (WES) on 154 unrelated sporadic ALS patients and 102 healthy controls from West China Hospital in Southwest China (Tab. S1, 2). We identified 404,047 variants, including 368,143 single-nucleotide variations (SNVs) and 35,904 insertions or deletions (Indels) that were predicted to alter protein-coding sequences. Among all the SNVs, 39,178 were classified as rare damaging variants (RDVs). Moreover, we identified 110 genes harboring 2 or more RDVs in the coding sequences that were not reported in ALS (Tab. S3). We then designed a custom panel of 288 candidate genes that included these 110 genes and 178

previously reported genes with various evidence of association (known ALS genes, GWAS, WES, functional, etc.) (Tab. S3). We analyzed the panel of genes in 392 ALS patients and 328 controls (including the case and control samples for the WES study) by ultra-deep targeted sequencing (Tab. S4). We identified 11,076 SNVs including 6335 rare non-synonymous variants, among which 3139 were predicted to be damaging (Tab. S5). 41 out of 392 (10.5%) patients carried one or more mutations in 18 known ALS genes, of which *SOD1* was the top mutated gene (Tab. S6). Given the low background rate of biallelic variants[55], and because the sample size required to characterize causative genes under a recessive model is much smaller than under a dominant model[56], we sought to prioritize biallelic RDVs (homozygotes or compound heterozygotes) in undescribed genes. Interestingly, in the 288 targeted genes, we identified three unrelated cases carrying a homozygous c.2099 T > C (p. Leu700Pro; rs782621196) variant in the *PCDHA9* gene (Fig. 1a–c). In addition, we found compound heterozygous variants in *LAMC3* (c.1330 C > T; p.R444C; c.2438 G > T; p.S813I) in one patient, and *RP1L1* (c.455 G > A, p.R152Q; c.199 G > A, p.E67K) in another patient. However, they were not recurrent in ALS patients and their correlations with the disease were not evident. Recessive *LAMC3* mutations were reported to cause malformations during the development of the occipital cortex, and most of them were loss-of-function (stop or frameshift) variants. Moreover, according to the ACMG/AMP classification guidelines, the c.2438 G > T (p.S813I) variant was determined to be benign/Likely benign, and c.1330 C > T (p.R444C) was determined to be VUS. In *RP1L1*, both c.455 G > A (p.R152Q) and c.199 G > A (p.E67K) were determined to be VUS.

The *PCDHα* gene cluster contains multiple variable exons that encode the six extracellular cadherin domains (ECD1–6), a transmembrane domain (TM), and three cluster-specific "constant" exons that encode a common intracellular domain (ICD). The pre-mRNA of *PCDHA9* is generated by cis-splicing joining variable exon 9 to three constant exons. In the current study, we identified 19 rare coding variants in this gene in cases, among which the L700P variant is the most frequent. More importantly, we detected homozygous L700P variant in 3 cases, but not in controls. The variant is located in the TM domain (Fig. 1b). Gene variant curation using ACMG/AMP guidelines (2015) suggests that it is a VUS (PM2_supporting +PP1_ Moderate +PS3_Moderate) variant. However, according to the ClinGen Bayesian classification framework (https://www.acgs.uk.com/quality/best-practice-guidelines/#VariantGuidelines), the variant got 5 points, which is very close to the criteria (6 points) for likely pathogenic variants. Currently, it is difficult to classify this variant as likely a pathogenic or pathogenic variant as DNA samples of the deceased parents of all 3 unrelated patients were not available and, thus, unable to get the evidence for in trans distribution of the variant in their parents. However, upon Sanger sequencing of additional ALS cases, we did not detect the homozygous variant in 548 Northern Chinese cases and 375 cases recruited from Guangdong province in South China. Searching the in-house WES databank of 14,154 samples, we identified 14 heterozygotes (0.00049), but no homozygotes. The gnomAD v2.1.1 WES database only reports 2 heterozygotes of this variant out of 9191 East Asians (MAF = 0.0001088), but none in other populations (http://gnomad-sg.org/variant/5-140230179-T-C?dataset=gnomad_r2_1). Similarly, the gnomAD v3.1.1 WGS database only reports 1 heterozygote out of 2592 East Asians (MAF = 0.0001929), and none in other populations (http://gnomad-sg.org/variant/5-140850594-T-C?dataset=gnomad_r3). Moreover, the homozygous L700P variant was not indexed in any database. In our study, the homozygous variant was identified in three cases from unrelated families, we postulated that it might be a rare mutation for ALS.

All the homozygous mutation carriers displayed relatively early onset (36–42 years) of limb weakness and survived less than 4 years. Since the onset, all patients manifested asymmetric and progressive weakness and atrophy of the extremities and bulbar muscles, with

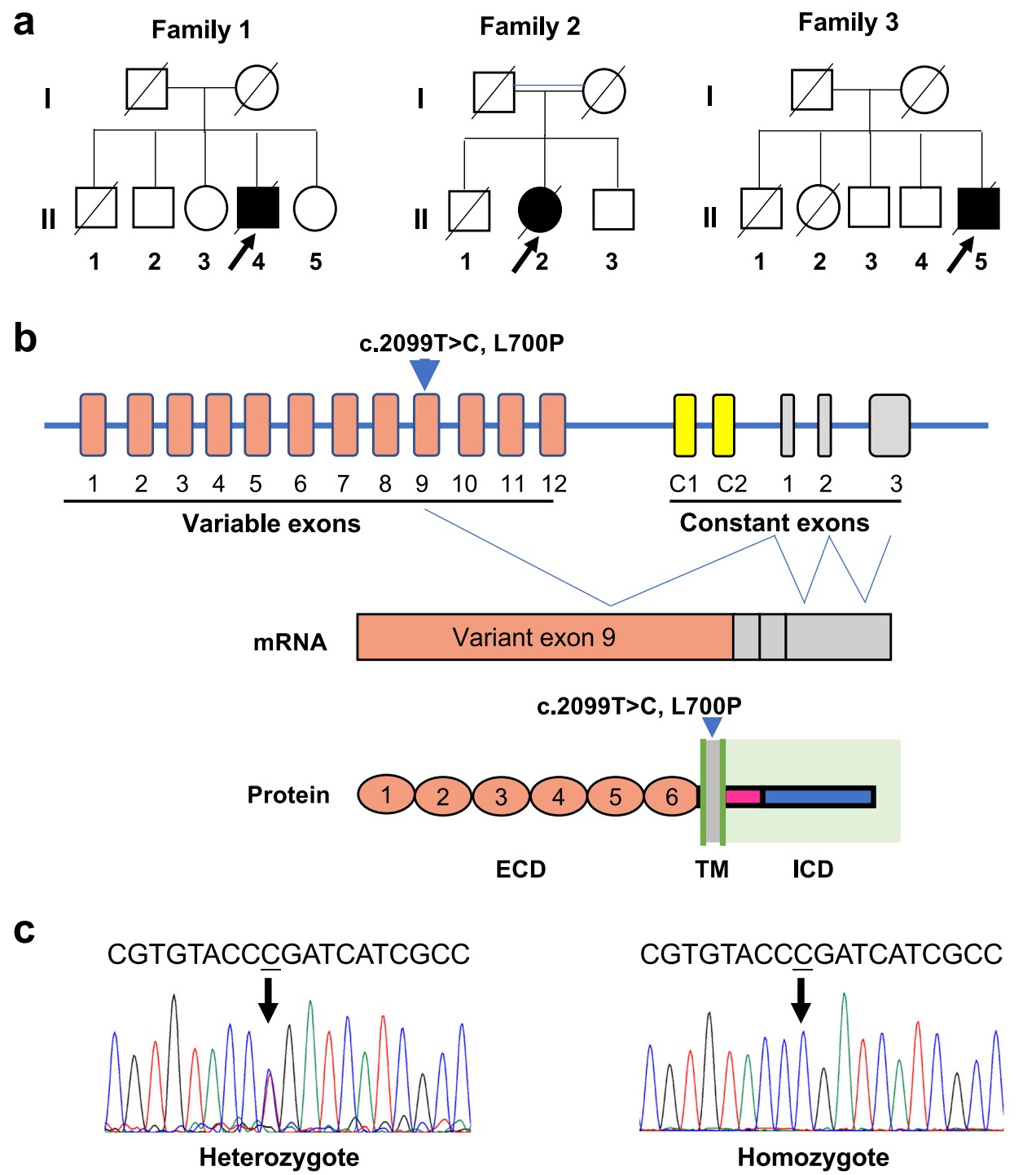

**Fig. 1 | Identification of the homozygous L700P mutation in *PCDHA9* in ALS patients. a** The families of the three unrelated patients harboring the homozygous L700P mutation. **b** Pcdha9 variable exons are joined via cis-splicing to three constant exons. Each variable exon encodes six EC domains (ECD), a transmembrane (TM) domain, and a short cytoplasmic extension. The constant exons encode the common intracellular domain (ICD). The mutation locates in the TM. **c** The sequence of the heterozygous and homozygous L700P mutation identified.

both upper motor neuron (UMN) and lower motor neuron (LMN) in at least three of the four regions of the body signs (bulbar, cervical, thoracic and lumbosacral). Electromyography (EMG) data displayed abnormalities including spontaneous activities and increased duration and amplitude of motor unit potential. According to the Gold Coast (GC) criteria for ALS[57], all these patients were diagnosed with definitive ALS. Genetically, they were tested to be negative for all the other ALS genes, and dominant or recessive variants of other unknown genes were excluded for association with the disease. The parents of the three homozygous carriers were either from consanguineous marriage (family 2) or families living in close proximity (family 1 and 3, <2 kilometers) (Tab. S7). Those patients' parents and other family members had not developed significant ALS symptoms (weakness, spinal or bulbar muscular atrophy).

### Generation of *Pcdhα9* point mutation and deletion mice
To build evidence for the pathogenicity of *PCDHA9* variants in ALS, we generated mutant mice based on this recurrent L700P mutation, which

is well conserved between mice and humans (Fig. S1A). We used CRISPR/Cas9 genome editing to target the *PCDHA9* orthologue in mice, generating homozygous mice carrying the missense mutation c.2185 T > C (p.L729P, corresponding to p.L700 in humans) (Mut, *Pcdhα9*[L729P/L729P]), or a single base deletion, c.2166delA (p.M723WfsTer119) (Del, *Pcdhα9*[M723Wfs/M723Wfs]), which caused frameshift and truncation of most of the transmembrane domain and entire intracellular domain (Fig. S1b, c). Quantitative real-time PCR analysis indicated a substantial decrease in *Pcdhα9* mRNA levels in Del mice, consistent with the effect of the nonsense-mediated decay, but not in Mut mice (Fig. S1d). The deletion mouse line was used to inspect whether the loss-of-function of PCDHA9 would result in phenotypes similar to the *Pcdhα9*[L729P] mutant.

## Both *Pcdha9* Mut and Del mice exhibit progressive motor function deficits

The body weights of both Mut and Del mice started to decline after reaching a peak around 7 months of age, and this decline became significant at 12 months (Fig. S1e). Most mice had curved spines (kyphosis) and were paralyzed at 14 months; paralysis usually started unilaterally, affecting limbs on one side at a time (Fig. 2a, Supplementary Movie 1). Some severely paralyzed mice (who lived in the SPF environment) died within 2 weeks of the onset of paralysis, resulting in a significantly lower survival rate than controls (Fig. 2b).

We then performed a series of behavioral tests on the mutant animals (in the clean grade room). The footprint test showed that the stride lengths in the stance of the hind limbs of both Mut and Del mice were slightly shorter as early as 4 months. The phenotype became obvious at 8 months and got more severe as they grew older (Fig. 2c, d). At 12 months of age, the mutant mice had severely abnormal gaits. Accompanying the abnormal gait, both forearm and four-limb grip strength became weaker at 12 months of age, but was not significant at 10 months or younger (Figs. 2e, S2a). Both Mut and Del mice showed significant deficits in motor function on the Rota-rod test at 12 months of age, while Del mice developed defects earlier than Mut mice at 10 months old (Figs. 2f, S2b). Moreover, both Mut and Del mice showed defects in limb coordination during the swimming test at 12 months of age (Figs. 2g, S2c). The mutant mice exhibited frequent abnormal claw movement and hindlimb over-extension when hung by their tail (Fig. 2h; Supplementary Movie 2). Interestingly, we noticed the differences in the Rota-rod and swimming tests, but not the grip force test, between heterozygous mut and wild-type male mice at the age of 16 months (Fig. S3). This implies that Pcdha9 heterozygous point mutation may have an effect on motor function in aged mice, although it is present at a much later stage.

Together, the above results demonstrate the progressive nature of the motor dysfunction phenotypes in Pcdha9 mutant mice. It is worth mentioning that Mut and Del mice used for the survival test under SPF conditions survived longer than those mice subjected to behavioral tests in clean-grade rooms. Thus, environmental factors may contribute to behavioral phenotypes.

## Neurogenic skeletal muscular atrophy in both Mut and Del mice

The progressive motor dysfunction of both Mut and Del mice led us to analyze their skeletal muscle, especially the gastrocnemius muscle, the largest and superficial skeletal muscle in the hind limb. Mut and Del mice exhibited severe muscle wasting. The size and weight of the muscle were remarkably reduced compared to controls at 13 months (Fig. 3a). We detected atrophied muscle cells with centrally localized nuclei, which was more obvious at 14 months than at 12 months (Figs. 3b, S4a). Atrophied muscle fibers indicated that muscle atrophy was neurogenic (Figs. 3b, S4a).

We performed EMG assessment and detected abundant positive sharp waves, fibrillations, and spontaneous and giant motor unit potential in muscles (including shoulder-deltoid muscle, trapezius dorsi muscle, and rectus femoris muscle) corresponding to multiple spinal cord segments in 13-month Mut mice, which were barely detected in WT littermates (Figs. 3c, S4b; Tab. S8). In addition, the spontaneous potentials detected were not symmetric between the two sides of each muscle. These data indicate that the extensive skeletal muscular atrophy and terminal-stage failure were due to neurogenic impairment, which are typical characteristics and essential diagnostic criteria for ALS in humans[58].

## Loss of motor neurons and abnormal neuromuscular junctions in the spinal cord

We went on to inspect the motor neurons in the ventral horn of the lumbar spinal cord (L3-L5 region). A significant loss of CHAT[+] motor neurons was detected in Mut mice at 12 months age, but no change at 10 month-old mice (Figs. 4a, S5a). Reduction of total neurons (NeuN[+]) in the ventral horn was also detected, although not as significant as that of CHAT[+] neurons (Fig. S5b). Meanwhile, the intensity of GFAP positive cells increased significantly in the vicinity of motor neurons (Fig. 4b), indicating reactive astrocytosis.

We then used RNAscope® probes to conduct in situ hybridization assays and found strong *Pcdhα9* expression in motor neurons (marked by CHAT probe) in the WT lumbar spinal cord (Fig. 4c), suggesting a specific function of Pcdhα9 in motor neurons.

To gain further insight into muscle atrophy, motor dysfunction, and paralysis, we examined the integrity of skeletal NMJs in the semitendinosus muscles from 13 month-old mice. We stained for α-bungarotoxin (α-BTX) to identify the acetylcholine receptor (AChR) cluster (motor endplate) in the postsynaptic site. Individual NMJ area was significantly reduced in both Mut and Del mice (Fig. S5c). Analysis of NMJs at presynaptic levels (NF-200) revealed significantly more presynaptic sites that were faint/weaker or even denervated in both Mut and Del mice (Figs. 4d, S5d). These abnormal NMJ structures could disrupt synaptic connectivity and cause defects in synaptic transmission. We also examined the NMJs of 6-month-old mice and found no significant differences in non-innervated and weakly AChR NMJs between WT and Mut mice at this stage (Fig. S5e).

We also traced up to the sciatic nerve by toluidine blue staining in 13 month-old mice. Mut mice had a significant increase in thinner axons (d < 20 pixels, 1 μm = 30 pixels) and a decrease in large-diameter axons (d > 70 pixels) (Fig. 4e). This indicates the degeneration and/or regeneration of axons, and fewer large caliber α-axons. A decrease of large axons, but not a defect in the myelination of fibers, was also detected by transmission electron microscopy at 13 months (Fig. S6a, b). Nevertheless, we did not observe any significant defects in myelination in the spinal cord, as shown by Black gold II staining (Fig. S6c).

More importantly, we observed the pathological feature of TDP-43 in the ventral spinal motor neurons of 12-month-old Mut mice. While TDP-43 aggregation was found in the cytoplasm of motor neurons from aged WT mice, we could detect distinct TDP-43 expression in the nuclei. However, in the Mut ventral spinal cord, TDP-43 was barely detectable in the motor neuron nuclei, whereas TDP-43 was abundantly accumulated in the cytoplasm (Fig. 4f). Of note, TDP-43 pathology in spinal motor neurons is a near-universal neuropathological hallmark of sALS[59,60].

The above results demonstrate that our old Mut and Del mice manifest typical ALS-like phenotypes including the progressive decline in survival and motor function caused by loss of spinal motor neurons (accompanied by astrocyte activation), significant muscle atrophy, and structural/functional abnormalities of NMJs. Motor neuron loss and NMJ abnormalities may contribute to the neurogenic atrophy of skeletal muscles (Fig. 4g).

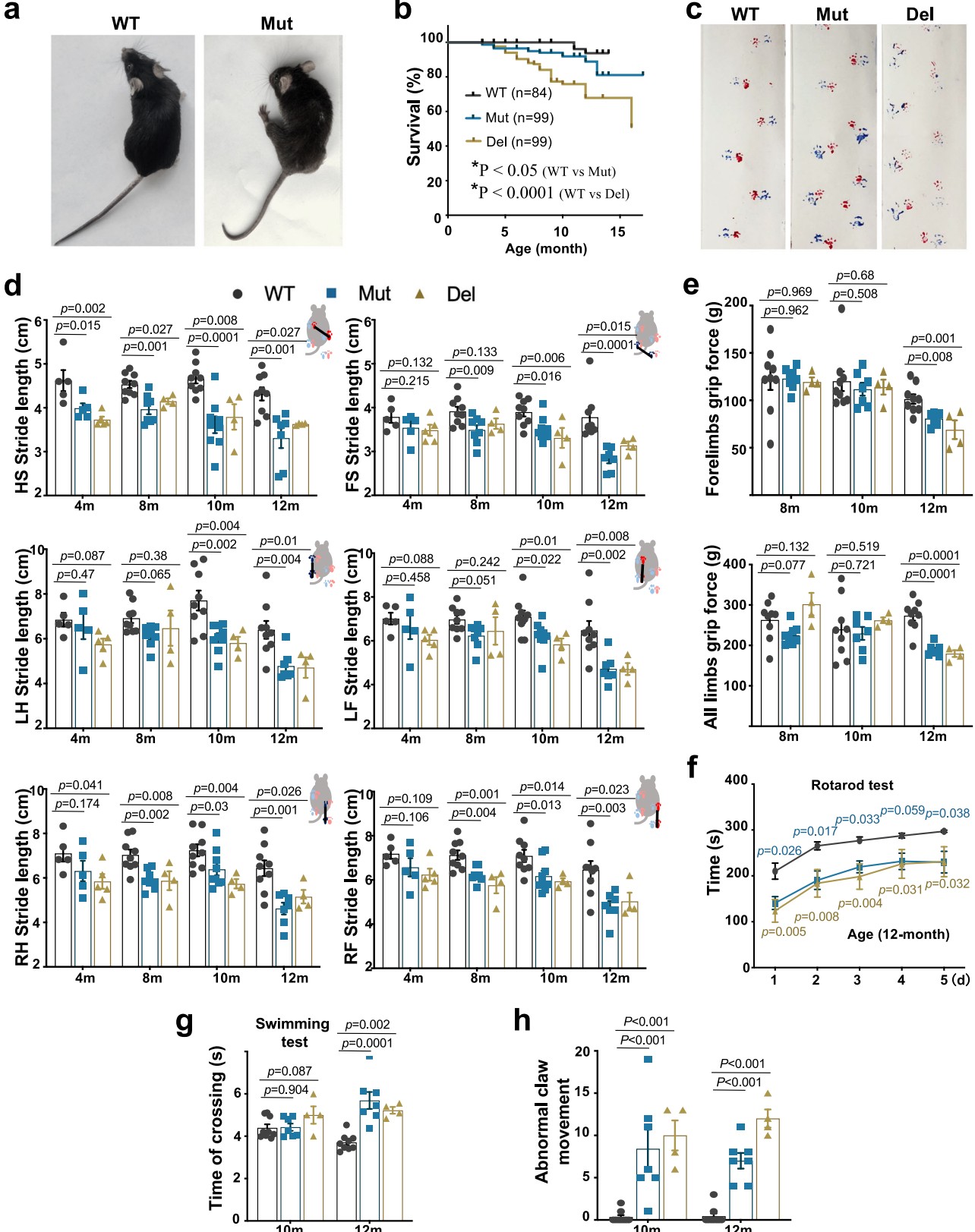

## Human PCDHA9 L700P mutation leads to its instability and dysfunction

To decipher the underlying pathogenesis in Mut mice, we performed a microdissection of the spinal ventral horn from 12-month-old mice and subsequent tandem mass tag (TMT)-labeled quantitative proteomic analysis to compare the protein levels between WT and Mut mice. The levels of 164 proteins changed significantly between WT and Mut mice. Gene Ontology (GO) analysis of those proteins indicated that the functions of (ion) transport, and cell or cell-cell adhesion were disturbed significantly (Fig. 5a, b). The other disturbed functions included response to cytokine, aging, mitophagy, nerve impulse transmission, epigenetic gene regulation and apoptosis, etc.

**Fig. 2 | Both *Pcdha9* Mut and Del mice exhibit progressive motor function deficits. a** Hemiplegia in the left hind limb of 14-month-old Mut mice. **b** Survival curves (WT, *n* = 84; Mut, *n* = 99; Del, *n* = 109. n represent biologically independent replicates). A Few Mut and Del mice started to die at 5 months old, and the death rate progressively worsened. **c**, **d** Abnormal footprints detected in both Mut and Del mice (12-month-old). Stride length (cm) and stance of limbs (left, right, hind limb and forelimb) tested by footprint tests as shown in (**c**) and analyzed at different ages (month) (HS: Hind limb stance, FS: Forelimb stance, LH: Left hind limb, LF: Left forelimb, RH: Right hind limb, RF: Right forelimb). The mouse cartoons showed the measurement diagram (red: forepaws; blue: hind paws). **e** Grip force

(gram) for limbs of the mice at 8-, 10- and 12-month old. **f** Rota rod test performed in five consecutive days. Time until falling was analyzed. *n* = 8 in WT, *n* = 5 in Mut group and *n* = 6 in Del group, *n* represents biologically independent replicates. *P* values with blue color represent p between WT and Mut; *P* values with brown color represent p between WT and Del. **g** Time crossing the swimming tank in the swimming test. **h** Statistical graph of the hind limb abnormal movement of claws in 10-month and 12-month old mice. All the mice used for behavior test are male mice. All data represent mean ± SEM. one-way ANOVA. For the behavior tests above, each group *n* ≥ 4 biologically independent replicates, the specific numbers are provided in source data. Source data are provided as a source data file.

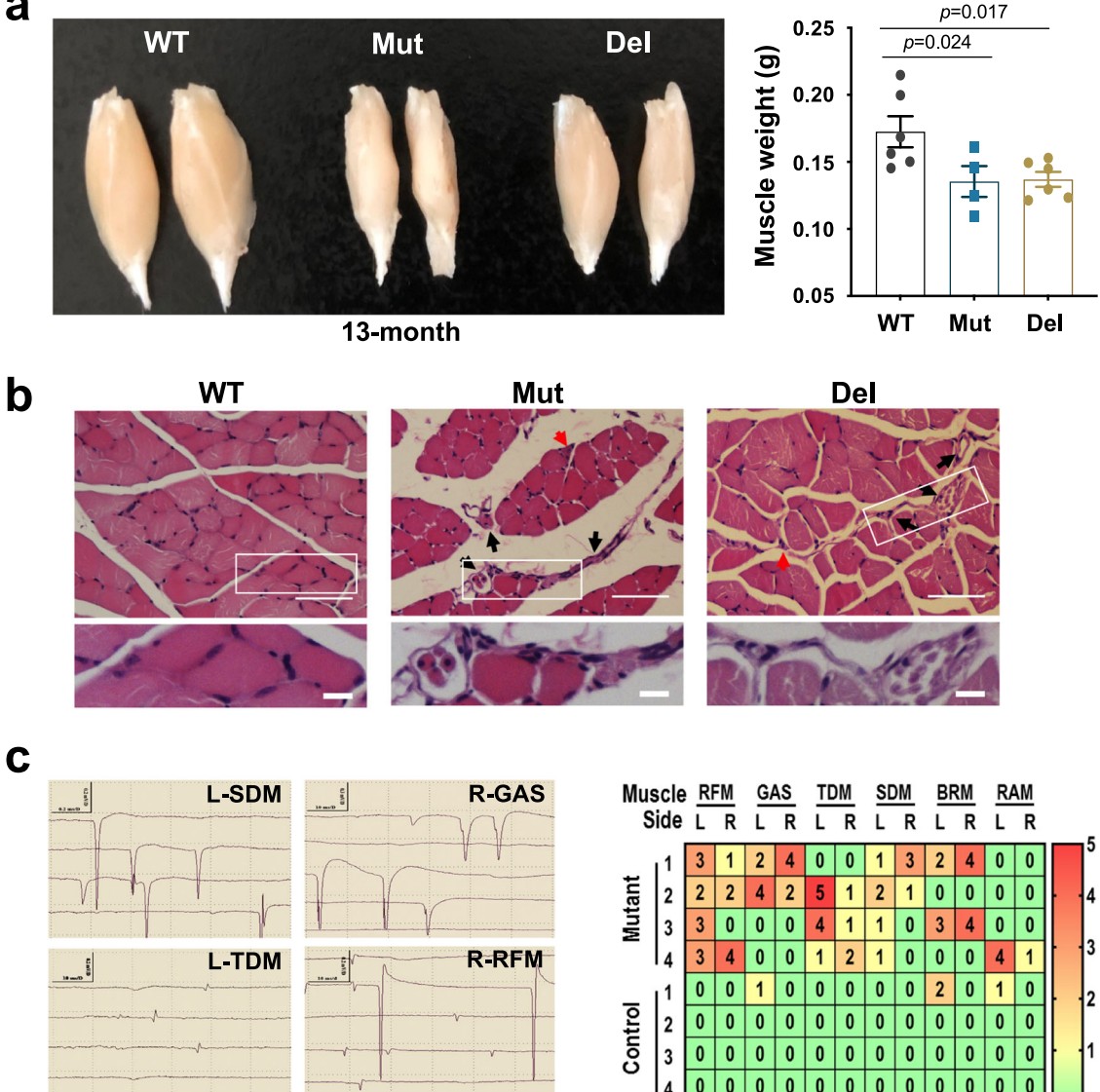

**Fig. 3 | Neurogenic skeletal muscular atrophy in both Mut and Del mice. a** The sizes and weights of gastrocnemius muscles from 13-month old mice. Data represent mean ± SEM. one-way ANOVA. *n* = 6 in WT and Del, *n* = 4 in Mut, *n* represents biologically independent replicates. **b** H&E staining of gastrocnemius muscles from 14-month-old mice. Black arrows: atrophied muscle fibers. Red arrows: centrally localized nuclei. White frame indicates the enlarged area. Scale bar: 50 μm in low power image, 10 μm in enlarged image. 3 biological repeats showed the same results. **c** Denervative potentials detected in multiple skeletal muscles in Mut mice.

Left panel: Positive sharp waves (PSWs) in the left shoulder-deltoid muscle (SDM) and the right gastrocnemius muscle (GAS), and fibrillation potentials (FPs) detected in the left trapezius dorsi muscle (TDM) and spontaneous motor unit potential and FPs in right rectus femoris muscle (RFM). Right panel: Heatmap of the distribution of denervative potentials (13-month-old mice, *n* = 4). Total number of PSWs and FPs for all four quadrants of a muscle were counted and indicated. L: left; R: right., BRM: biceps brachii muscle; RAM: rectus abdoininis muscle. Source data are provided as a source data file.

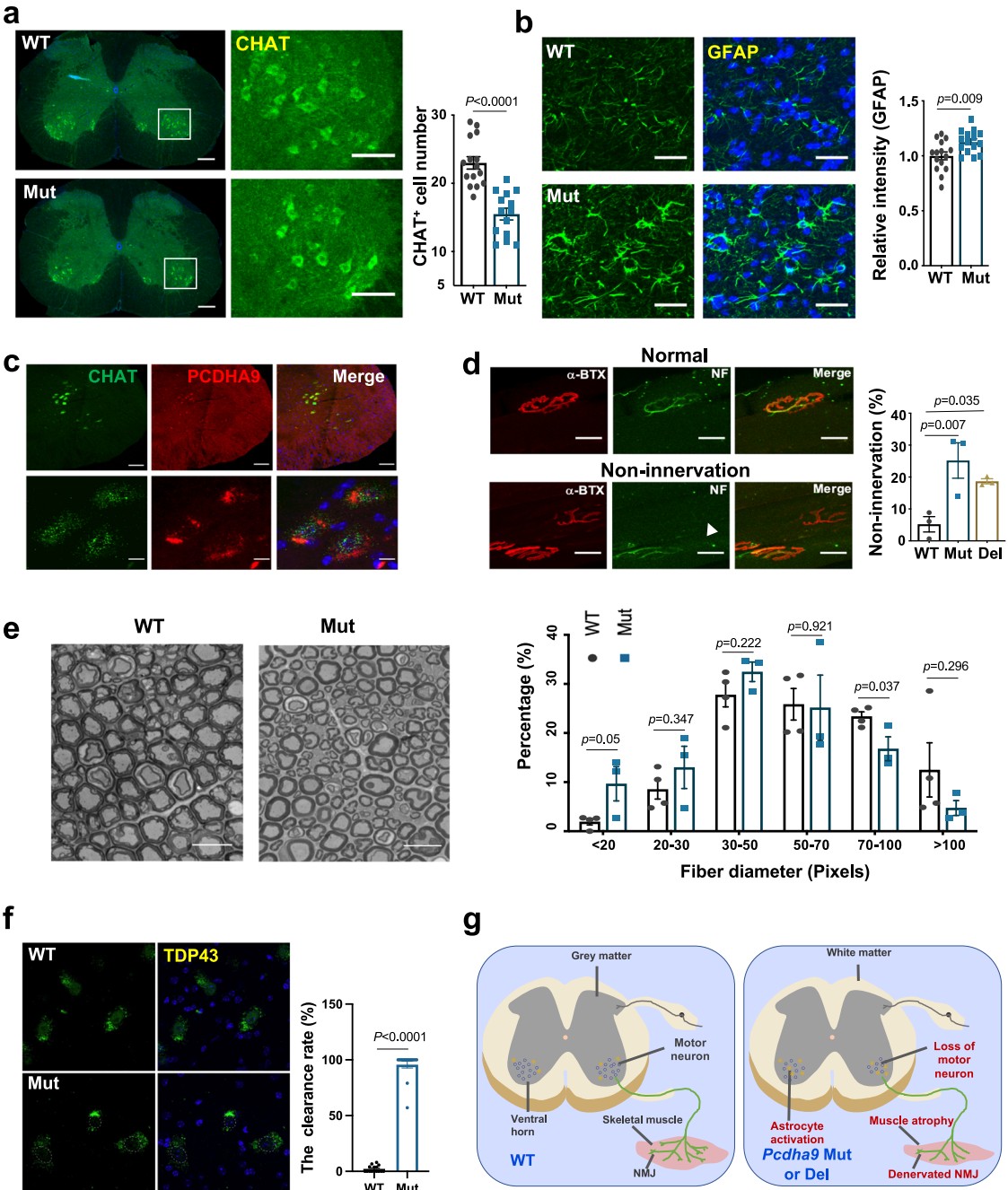

**Fig. 4 | Motor neuron degeneration in spinal cord and abnormal NMJs.**
**a**, **b** Confocal images and statistical analyses of choline ChAT-positive motor neurons (**a**) and GFAP⁺ cells (**b**) in ventral horn of the lumbar spinal cord from 12-month-old mice. White frame in (**a**) indicates the enlarged area. *N* = 3 mice; *n* = 15 slice, *t*-test. Scale bar in (**a**) whole spinal cord images: 200 μm; enlarged images: 100 μm. scale bar in (**b**): 25 μm. *t*-test. Data represent mean ± SEM. **c** RNAscope analysis of Pcdhα9 expression in 13-month wildtype mice spinal cord. CHAT: green; Pcdhα9: red. scale bar in low power images:100 μm; enlarged images: 10 μm. 3 biological repeats showed the same results. **d** Images of normally innervated and non-innervated NMJs and analyses of non-innervated NMJs. NMJs were stained with α-BTX (red) and neurofilament-200 (NF-200, green). White arrow indicated non-innervated AChR. Scale bar: 30 μm. *n* = 3 mice for each group. NMJs:

*n* = 452 in WT, *n* = 736 in Mut, *n* = 620 in Del mice. one-way ANOVA. Data represent mean ± SEM. **e** Toluidine blue staining of sciatic nerve (13-month mice) and analyses of fibers with different diameters. Scale bar: 100 μm. 1 μm = 30 pixels. *n* = 6 slices from 3 mice, one-way ANOVA. Data represent mean ± SEM. **f** TDP-43 images and statistical analysis in 12-month WT and Mut mice. Percentage of motor neuron with nuclei clearance of TDP-43 was analyzed. *n* = 15 slices from 3 mice. *t*-test. Data represent mean ± SEM. **g** A schematic model of typical ALS pathology in *Pcdha9* Mut and Del mice. Loss of motor neuron accompanied by activation of astrocyte in ventral spinal cord, skeletal muscle atrophy and NMJ denervation are detected in *Pcdha9* Mut and Del mice. Those defects lead to motor dysfunction. Source data are provided as a source data file.

Because *PCDHA9* is a member of the *PCDHα* family which plays a role in cell-cell adhesion during neural circuit assembly, we therefore explored the effect of the L700P mutation on PCDHA9, we expressed WT human PCDHA9 (hPCDHA9) or its mutant (hPCDHA 9

L700P) in HEK293 cells. The protein level of the hPCDHA9 mutant was lower than that of WT, while the mRNA levels were similar (Fig. 6a–c). We also tested their expression on the plasma membrane by staining live cells with an hPCDHA9 antibody, and found that the

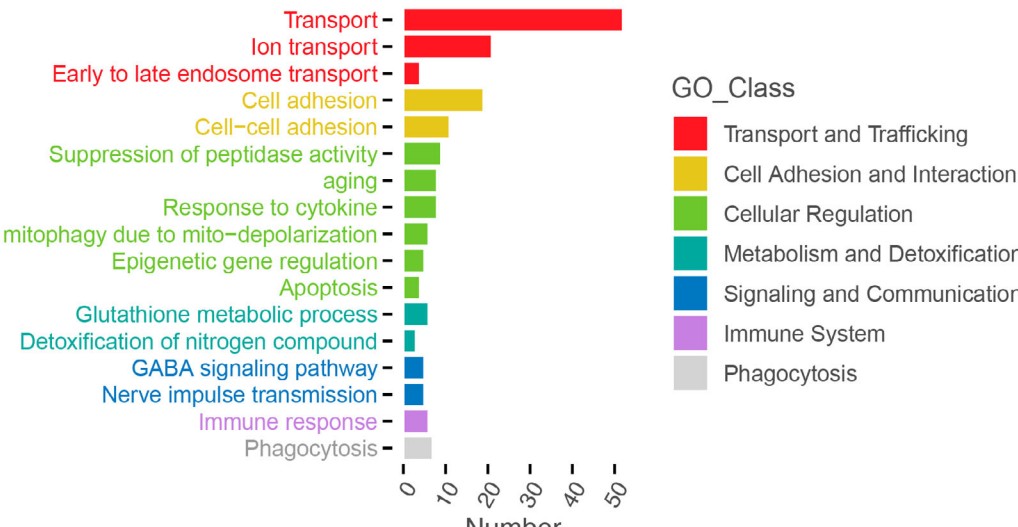

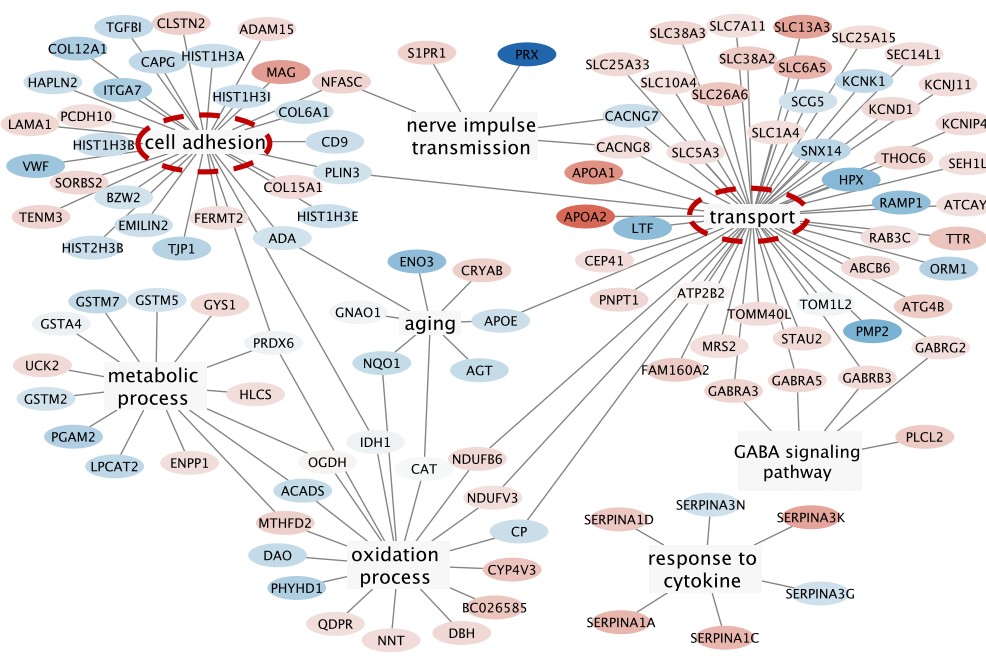

**Fig. 5 | Tandem mass tag (TMT)- labeled quantitative proteomic analysis of the ventral horn of spinal cord from Mut and littermate control mice. a** GO analysis of quantitative proteomic data of the ventral horn of L3–L5 spinal cord from the 12-month-old WT and Mut littermates. **b** Network graph of the significantly changed proteins with their functions. The graph was generated by the Cytoscape software. Red color indicated the increased expression levels in Mut mice compared with WT mice, while blue color indicated the decreased ones. The depth of the color represents the extent of changes. Log$_2$ FD > 1.25 and Log$_2$ FD < 0.75.

signal was also much weaker in hPCDHA9 L700P expressing cells (Fig. 6b). When cells were treated with protein synthesis inhibitor cycloheximide (CHX), we detected much shorter half-life of mutant hPCDHA9 (Fig. 6d). Meanwhile, treatment of cells with the proteasome inhibitor MG132 resulted in significantly elevated expression of mutant hPCDHA9 (Fig. 6e). These results suggest that L700P mutation of human PCDHA9 leads to decreased stability and increased degradation.

Overexpression of PCDHα-C2 and PCDH-γ-C5 proteins were reported to interact and inhibit the activity of focal adhesion kinase (FAK) family members PYK2 and FAK, through their C-terminal which are shared in the family[61]. Deletion of the *Pcdh-γ* family in mice or the *Pcdh-α* family in zebrafish was shown to induce neuronal apoptosis[62,63]. Neuronal death caused by deletion of the whole *Pcdh-γ* family is likely due to the upregulation of PYK2 and FAK activity in the mouse spinal cord[61]. We therefore co-expressed hPCDHA9 and FAK or PYK2 in HEK293 cells and validated the interaction of hPCDHA9 with PYK2 (Fig. 6f). We found that phosphorylation levels of both FAK and PYK2 were substantially inhibited by WT hPCDHA9, but not by mutant hPCDHA9 (Fig. 6g, h). Importantly, phosphorylation levels of FAK were significantly increased, and the phosphorylation levels of PYK2 also trended upward in the spinal cords of 12 month-old Mut mice

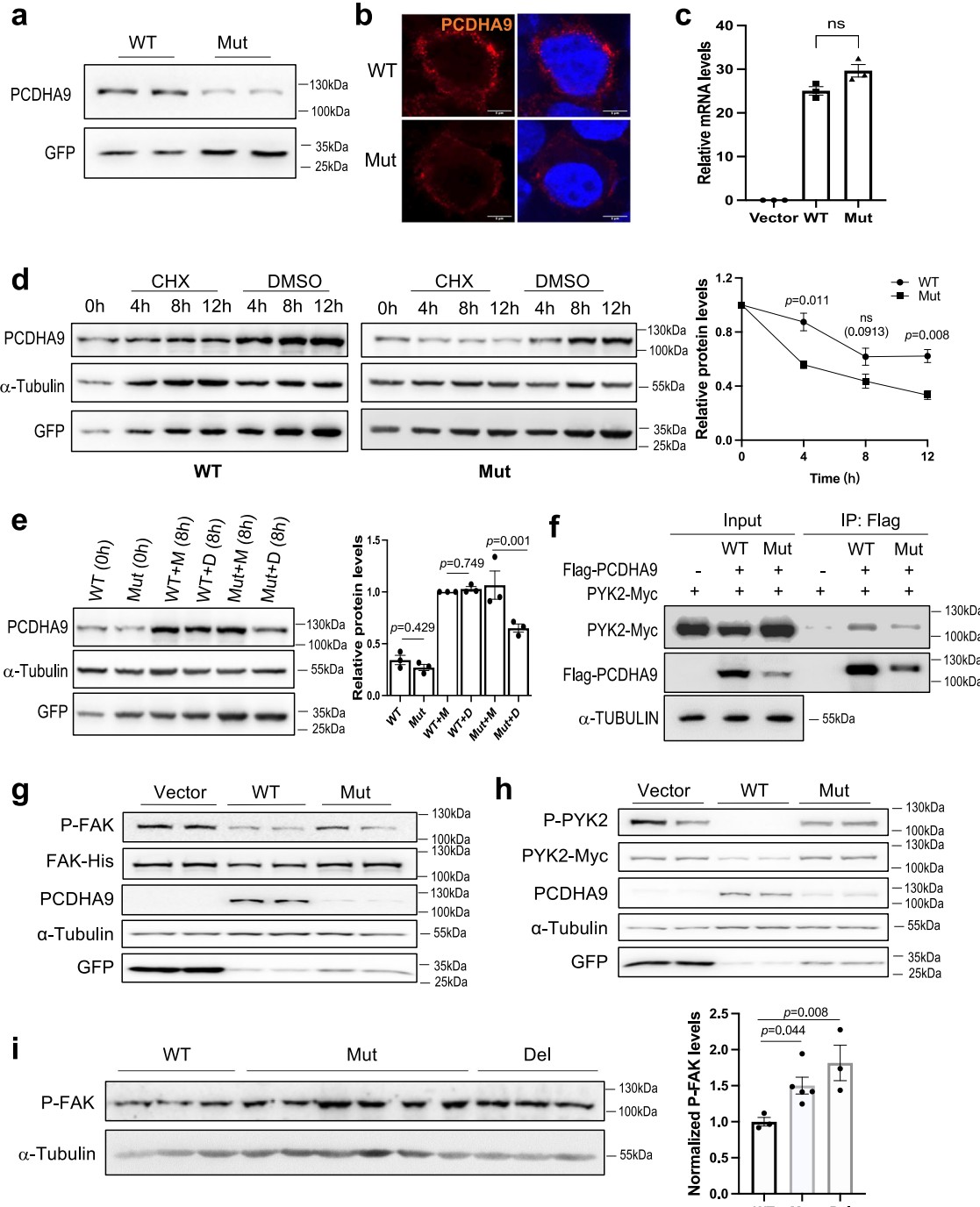

**Fig. 6 | Mutation of hPCDHA9 results in its instability and abnormal FAK phosphorylation.** WT or mutant hPCDHA9 in pCMS.EGFP vector were transfected in HEK293 cells as indicated (**a**–**h**). **a** Western blot of WT and mutant hPCDHA9 levels. GFP was used as transfection efficiency controls. **b** Confocal images of hPCDHA9 expression on the plasma membrane. Scale bar: 5 μm. 3 independent repeats showed similar results in (**a**) and (**b**). **c** Relative mRNA levels of transfected WT and mutant hPCDHA9 analyzed by RT-PCR and normalized by GAPDH. $n = 3$ independent replicates. one-way ANOVA. Data represent mean ± SEM. **d** Expression of WT or mutant hPCDHA9 treated with/without CHX for different time. GFP and α-TUBULIN were used as transfection efficiency and loading controls. one-way ANOVA. Data represent mean ± SEM. $n = 3$ independent replicates. **e** Expression of WT and mutant hPCDHA9 in cells treated with/without the proteasome inhibitor

MG132 for 8 h. Data represent mean ± SEM. one-way ANOVA. $n = 3$ independent replicates. (M: MG132, D: DMSO). **f** Anti-Flag immunoprecipitation was performed with cells expressing the indicated constructs. The input and immunoprecipitated proteins were detected by Western blots. **g**, **h** Phosphorylation and total protein levels of FAK and PYK2 in 293 cells transfected with WT or mutant hPCDHA9 and FAK or PYK2. GFP and α-Tubulin were used as transfection efficiency and loading controls, respectively. 3 independent repeats showed similar results in (**f**–**h**). **i** Phosphorylation levels of FAK in the spinal cord of 12-month old WT, Mut and Del mice. Statistic analysis of P-FAK levels was performed by normalizing with α-Tubulin. $n = 3$ in WT and Del groups and $n = 6$ in Mut group. n represents the biologically independent replicates. The data represent mean ± SEM. one-way ANOVA. All uncropped blots are provided in Source Data file.

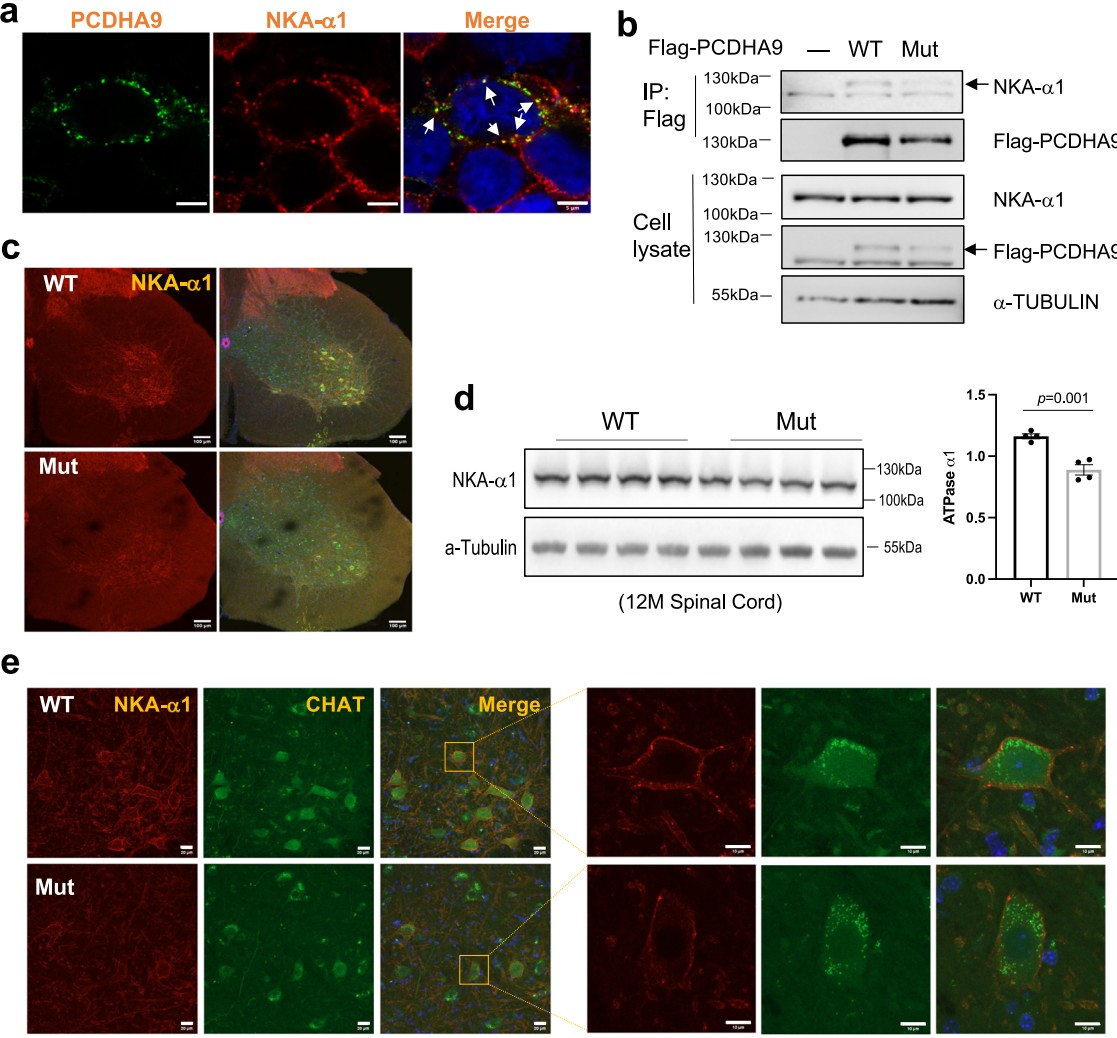

**Fig. 7 | Na⁺/K⁺ ATPase-α1 expression in spinal motor neurons is significantly downregulated in *Pcdha9* mutant mice. a** Co-localization (white arrow) of over-expressed hPCDHA9 with endogenous Na⁺/K⁺ ATPase-α1 on the plasma membrane in 293 cells. Scale bar: 5 µm. **b** Endogenous Na⁺/K⁺ ATPase-α1 interacts with over-expressed WT and mutant hPCDHA9 in 293 cells. Anti-FLAG immunoprecipitation was performed. The input and immunoprecipitated proteins were detected by Western blots. Uncropped blots are shown in Source Data. 3 independent repeats showed similar results in (**a**) and (**b**). **c** Confocal images of Na⁺/K⁺ ATPase-α1 expression pattern in the lumbar spinal cord of 12-month-old mice (crossed with HB9-GFP mice). The signal of Na⁺/K⁺ ATPase-α1 is strong in the motor neurons. scale bar: 100 µm. Na⁺/K⁺ ATPase-α1 (red). **d** Expression of Na⁺/K⁺ ATPase-α1 in 12-month-old mice lumbar spinal cord. *t*-test. The statistic data represent mean ± SEM. *n* = 4 biologically independent replicates. All uncropped blots were provided in Source Data. **e** Enlarged images of Na⁺/K⁺ ATPase-α1 in ventral horn of spinal cord (L3–L5), the signal in motor neurons (marked by CHAT) is decreased in Mut mice. *n* = 3 each in WT and Mut mice, n represent the biological repeats. Scale bar for low power images were 20 µm and for enlarged images were 10 µm. Source data are provided as a source data file.

(Figs. 6i, S7). These results indicate that aberrant activation of FAK and PYK2 might contribute to neuronal loss in the spinal cord of our Mut mice.

### *PCDHA9* mutation causes reduced expression of Na⁺/K⁺ ATPase-α1 in spinal motor neurons

Since (ion) transport is the most significantly disturbed function in the above GO analysis, we inspected the expression of membrane proteins involved in ion transport. Na⁺/K⁺ ATPase-α1 (NKA-α1), a member of the Na⁺/K⁺ ATPase-α family, is essential for the maintenance of the sodium and potassium gradient across neuronal membranes and cell survival[64]. We used NKA-α1 as the plasma membrane marker and found that it co-localized with hPCDHA9 (Fig. 7a). This suggests that hPCDHA9 interacts with NKA-α1. To test this, we performed co-immunoprecipitation of Flag-hPCDHA9 expressed in HEK293 cells and detected the presence of endogenous NKA-α1 in the immunocomplex (Fig. 7b). We then inspected the expression of NKA-α1 in mouse spinal cord. As shown in Fig. 7c, e, it is selectively and highly expressed in the

HB9-GFP and CHAT-positive motor neurons. When examining NKA-α1 protein levels in the spinal cords of 12-month-old mice, we found that it was significantly lower in Mut mice (Fig. 7d). Meanwhile, immunos-taining revealed lower NKA-α1 levels in the motor neurons of Mut mice (Fig. 7c, e).

The functional core of NKA-α is the α/β-heterodimer, which often associates with a small transmembrane protein of the FXYD family to form a heterotrimeric complex (α/β/FXYD)[65]. We examined the expression of Na⁺/K⁺ ATPase α3 and β1 (NKA-α3 and β1) and found that they were not significantly altered in the spinal cord of 12 month-old Mut mice (Fig. S8a, b). The above results indicate that reduced NKA-α1 expression caused by mutation of *PCDHA9* could make spinal motor neurons more vulnerable.

### Single nucleus RNA-seq and ATAC-seq to define defects in *Pcdha9* mutant mice

To decipher the underlying pathogenesis and define the different programs of gene expression before the onset of motor dysfunction,

we again micro-dissected the ventral horn from the spinal cords of Mut mice and their WT littermate mice and performed single nucleus RNA sequencing (snRNA-seq) and single nucleus assay for transposase-accessible chromatin sequencing (snATAC-seq) of cells from 9 month-old mice. After quality control, we obtained 21756 cells in snRNA-seq and 14095 cells in snATAC-seq respectively, and divided these cells into 5 clusters (Figs. 8a, S9a–g, S10a). Each cluster was defined by well-known marker genes based on the mRNA levels in snRNA-seq datasets or gene scores in snATAC-seq datasets (Figs. 8a, b, S10a, b, S11, S12). We performed both t-distributed stochastic neighbor embedding (t-SNE)[66] analysis and uniform manifold approximation and projection (UMAP)[67] to visualize identified cells including neuron, oligodendrocyte, astrocyte, microglia and oligodendrocyte progenitor cell (OPC) (Figs. 8a, S10a).

In the Mut ventral horn neuron cluster, 62 genes were significantly downregulated and 57 genes were significantly upregulated at the transcriptional level in the snRNA-seq datasets (Fig. 8c). The representative significantly enriched GO categories among downregulated genes are synapse organization, regulation of ion transmembrane transport, potassium ion transport, regulation of membrane potential and cell junction assembly (Fig. 8c). The upregulated genes were enriched in myelination and organic acid or fatty acid biosynthetic process that implicate the compensatory effects in the Mut spinal neurons. Of those differentially expressed genes (DEGs), 17.5% are associated with ion transport, 12.5% with synapse and 8.33% with cell adhesion (Fig. S10c). Interestingly, 17.5% of those genes were reported in studies of neurodegenerative disease, including ALS (*Fggy*, *Fgf1*, *Ugt8a* and *Actb*)[68–71] (Fig. S10c). Among those DEGs, *Ptprt* and *Plekhg1* encode proteins that interact with the PCDHA family binding protein Fyn[72,73].

Based on chromatin accessibility in the snATAC-seq datasets, we identified 5692 (false discovery rate [FDR] <0.05) differentially accessible regions (DARs) in the neuron cluster (WT and Mut) with 4696 DARs that were less accessible and 933 DARs that were more accessible. The enrichment of biological functions in the flanking genes of the DARs showed that, in addition to those shown in DEGs in Fig. 6c, different signaling pathways, muscle system process, axonogenesis, dendrite development, response to oxidative stress, regulation of translation, were affected obviously (Fig. S10d, e).

According to a previous report, 154 proteins including PCDHA9 interact with PCDHγ in the mouse brain[74]. Therefore, those proteins might interact with PCDHA9. Mapping these proteins with our DEG-encoded proteins in the neuron cluster led to the identification of 4 overlapping proteins, HSPA8, NEFM, ACTB and LSAMP, in which LSAMP is a neural adhesion protein[75] (Fig. 8d). We also mapped these 154 proteins with flanking genes of DARs encoded proteins in our snATAC-seq data and found 43 overlapping proteins (marked as circles, Fig. 8d). According to our snRNA-seq data, HSPA8, ACTB and LSAMP were DEG-encoded proteins, but not the other 40 proteins. This might be due to the limited sequencing depth of snRNA-seq or other reasons. We predicted genes that might be related to these 43 genes through geneMANIA (http://genemania.org/)[76]. From them, we identified 6 additional genes (marked as diamonds, Fig. 8d) that also had significant chromatin accessibility differences in our snATAC-seq data. All these 49 (43 + 6) proteins showed extensive interactions with each other. GO analysis revealed that 38 proteins are located on the plasma membrane, synapse or cell junction, and 28 of them have transport function. This suggests that mutation in PCDHA9 affects transport between cells or synapses. Among these 49 proteins, only RXRA is a transcription factor, and its downstream target genes in Mut mice predicted by gene regulatory networks (GRNs) also regulate transmembrane ion transport and synapse organization (Fig. S10f).

Notably, Fxyd2, a specific subunit of NKA-α1 heterotrimeric complex[77], is also present in the mapping result (labeled in blue color, Fig. 8d). Thus, the potential interaction between FXYD2 and PCDHA9 complements our above results showing the interaction between NKA-α1 and PCDHA9 (Fig. 7a, b). The differences in chromatin accessibility in *Fxyd2* and *Atp1a1* (encoding NKA-α1) between WT and Mut mice snATAC-seq datasets highlight the effect of *Pcdha9* mutation on the function of ion transport (Figs. 8d, S10d).

Taken together, the above analyses indicate that the mutation of *Pcdha9* leads to dysregulation of genes involved in neuronal cell adhesion, ion transport, and synapse function, as well as neurodegeneration.

## Alterations in gene regulatory networks and intercellular communications in Mut mice

We generated potential gene regulatory networks (GRNs) by linking distal cis-regulatory elements after integrating snRNA-seq and snATAC-seq datasets to individual genes according to our previous report[78] to identify candidate transcriptional factors (TFs) and regulators (Fig. S9h, i). Among all the regulatory pairs captured, we selected those upstream TFs and downstream genes that were highly expressed in each cluster and visualized by Cytoscape (Fig. 8e). Each cell type displayed specific and shared regulatory networks in WT and Mut mice. GRNs exhibited obviously different complexity between WT and Mut cell subtypes (Fig. 8e). Importantly, the GRNs indicate that *Rxra*, the only TF found in the above mapping result (Fig. 8d) had much more candidate downstream genes in Mut mice compared to WT mice. One of them, the DEG *Malat1*, had specific opening chromatin peaks in the Mut neuron cluster, and the predicted *Rxra* binding sites were mapped to these peaks (Fig. S10g). Thus, *Rxra* might regulate the specific open region of *Malat1* in Mut mice and upregulate the expression of *Malat1* in neuron cluster.

Expression of *MALAT1* was significantly upregulated in the brains of PD and FTLD-TDP patients, and affected multiple pathways, including apoptosis, autophagy, and neuro-inflammation[79–83]. Because *Malat1* is highly conserved between humans and mice, and human database is more comprehensive and informative, we used LncBase v.3 (https://diana.e-ce.uth.gr/lncbasev3/home)[84] in the human database to predict the downstream miRNAs regulated by *Malat1*. We obtained 412 miRNAs, of which 42 were reported as neuron axon-localized miRNAs (Fig. 8f). We searched for target genes of these miRNAs predicted by miRwalk (http://mirwalk.umm.uni-heidelberg.de/)[85]. We found 35 of them were upregulated in our snRNA-seq datasets, including *Fth1*, *Actb*, *Elavl2*, *Fgf1*, *Mal* that were reported in studies of degenerative diseases[68,71,86–89], as well as *Plekhg1* which encodes a protein associated with PCDHA9[73]. Notably, *Fgf1*, *Act* and *Elavl2* are upregulated in motor neurons of ALS patients or animal models[71,88,89]. Thus, the upregulation of *Malat1* and its downstream genes might contribute to neurodegeneration in Mut mice.

Finally, to map the complex ligand–receptor interactions, we inferred all possible intercellular communication among different cell types through CellPhoneDB v.3.0 in snRNA-seq datasets[90]. The total normalized number of ligand-receptor interactions within or between different cell types was increased in Mut mice (Fig. 8g). The ligand-receptor interactions detected specifically in Mut mice are shown in Fig. 8h and Fig. S9h. Coincident with our result, increased expression of IGFR and IGF1R has been reported in the frontal lobe of FTLD postmortem[91]; the EGF/EGFR is activated at the neuritic plaques in Alzheimer's patients[92]; expression of EFNA3 and EPHA7 are elevated in neurodegenerative diseases or spinal cord injury, and EPHA7 is predominantly expressed in motor neurons[93,94]; EPHA3 deletion or EPHA4 inhibition delays the ALS process[95,96]; TGF-β2 has a tendency to increase in the lumbar spinal cord of ALS postmortem and potentially accelerates the neuronal cell death in AD[97–99]; WNT5A is upregulated as

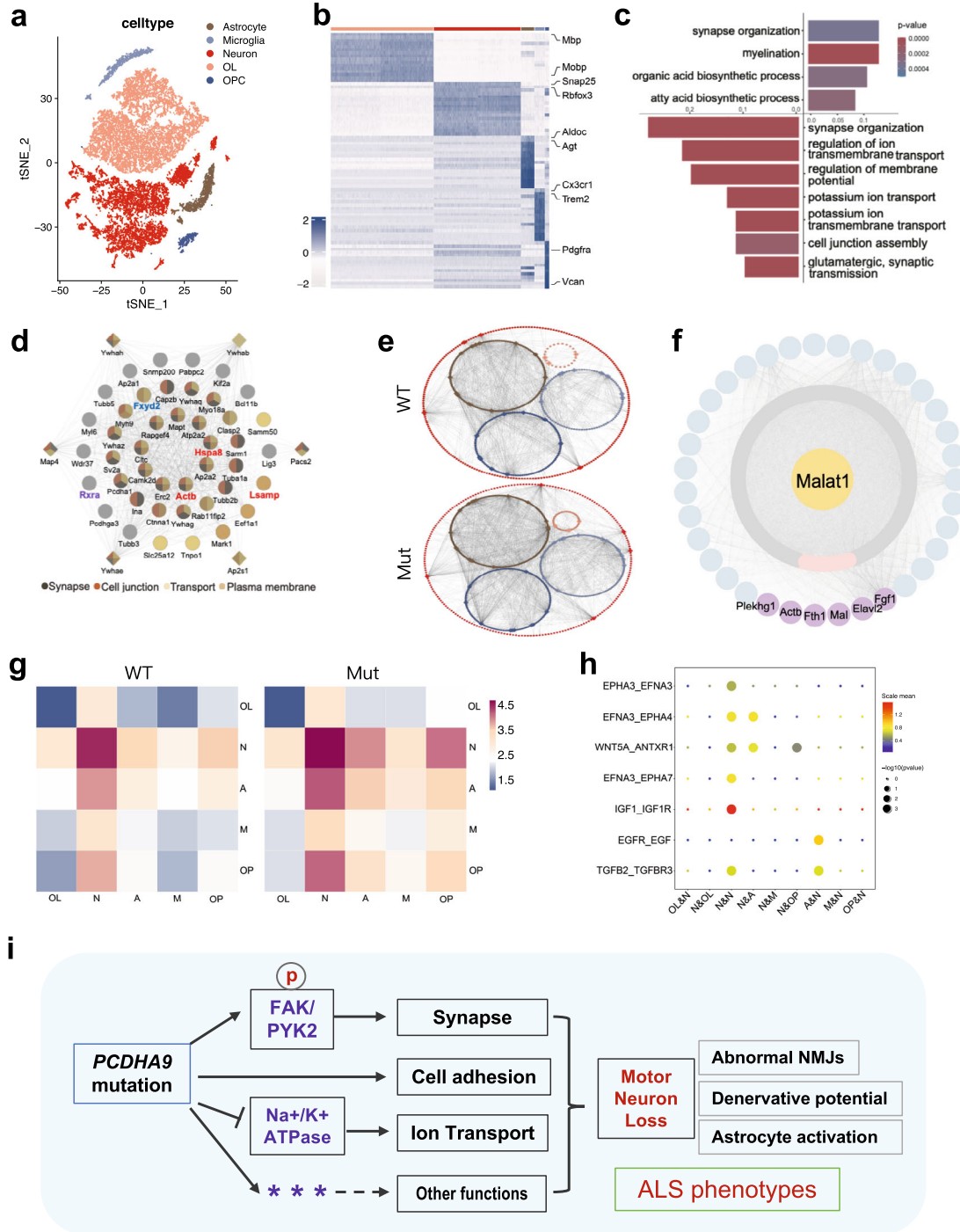

**Fig. 8 | SnRNA-seq and snATAC-seq analysis of spinal cord ventral horns of 9-month-old mice. a** Cell clustering of integrated snRNA-seq data visualized by t-SNE plot. Clusters were grouped and annotated according to the expression of marker genes. **b** Heatmap describing marker genes enriched in 5 cell types. **c** GO of significant DEGs in neuron, determined by Wilcoxon test, were selected for GO analysis. Upregulated genes in Mut mouse (top), downregulated genes (bottom). **d** PPI analysis of proteins interact with PCDHA9 (circle) and their predicted associated proteins (diamond), that are also encoded by genes associated with significant DARs in snATAC-seq. Colors represent different GO terms. Proteins marked in red are also DEGs (Hspa8, Lsamp and Actb), TF Rxra is marked in purple and Fxyd2 that interacts with NKA-α1 is marked in blue. Network was generated by STRING tool. **e** GRN generated based on integration of snRNA-seq and snATAC-seq. **f** Malat1 was

predicted to modulate the 35 upregulated genes in snRNA-seq neuron cluster (the outermost ring) through miRNAs (the middle ring). The miRNAs marked with pink color represent neuron axon-localized miRNAs and the genes in the outermost ring marked with purple are degenerative diseases associated genes and the one related to Fyn (Plekhg1). **g** Heatmap depicting the total normalized number of ligand-receptor interactions between cell types in WT and Mut snRNA-seq data obtained with CellPhoneDB v.3.0. **h** Potential neurodegeneration diseases associated ligand-receptor pairs linking neuron to other cell types shown as scaled mean, as calculated by CellphoneDB. The analysis was based on significance level of $p < 0.5$. Short name in (**g**, **h**): (OL: oligodendrocyte. N: neuron. A: astrocyte. M: microglia. OP: oligodendrocyte progenitor cell). **i** Schematic diagram of the mechanism by which mutations in PCDHA9 lead to ALS. Source data are provided as a source data file.

a regulator of neuroinflammation in the brain of an AD mouse model[100]; Antxr1 was reported as biomarker of AD[101,102]. The changes in those ligand-receptor interactions might either contribute to neuronal degeneration or compensate for neuron survival. Thus, the above analyses provide an overview of GRNs and intercellular communications that are potentially associated with degenerative diseases, including ALS, in Mut mice.

## Discussion

Recent progress in ALS genetics and the discovery of ALS genes presents an opportunity to generate models to explore the pathogenesis of the disease and identify therapeutic targets shared between distinct forms of ALS. Through genetic screening of ALS patients, we identified PCDHA9 as a potential ALS gene and verified from the mouse models (Fig. 4g). Mechanically, we demonstrated that PCDHA9 mutation leads to its reduced expression on the cell membrane and its instability, which will lead to a decrease in cell-cell adhesion. Meanwhile, the function of its interacting proteins would also be affected, including excessive activation of FAK and/or PYK2 which may lead to the degeneration of synapse and reduced level of NKA-α1, which is essential for ion transport. Accordingly, our multi-omics analysis revealed disturbed networks or signaling pathways involved in cell/cell-cell adhesion, ion transport, synapse organization, and neuronal survival in aged mutant mice. Our findings on many other potential signal pathways and gene regulatory networks would provide more clues for future research (Fig. 8i).

Recent next-generation sequencing studies have identified more genes associated with ALS, including *MATR3*, *CHCHD10*, *TBK1*, etc. These genes encode proteins with diverse biological functions, such as RNA metabolism, global protein homeostasis, or axonal transport dynamics[20]. However, the association of these genes with ALS has yet to be examined in animal models except for *MATR3*. We applied whole-exome sequencing combined with ultra-deep targeted sequencing of sALS cases and identified a homozygous rare damaging variant in *PCDHA9* in three unrelated sporadic cases. Heterozygotes for this variant are very rare (MAF < 0.0005) in gnomeAD and in-house databases, and homozygotes have not been reported in any databases. The homozygous variant also matched with two other carriers in the online GeneMatcher website: https://genematcher.org/reports/events?submissionID=58850&eventType=3. The carriers were both Western children with ID/ASD/Rett-like syndrome (Match Identifier: 19S0538) and Benign infantile epilepsy, respectively. Although the phenotypes were different from those in our study, the matching results implicate that the homozygous L700P variants in PCDHA9 are pathogenic and may be associated with different phenotypes in children and adults. Moreover, the homozygous variant was confined to Sichuan province in Southwest China, but not other Southern and Northern provinces, suggesting it may be a geographically associated variant. This finding expands the genetic spectrum of ALS and further confirms the high genetic heterogeneity among different ethnic populations.

As more candidate ALS genes are being discovered, the challenge is to determine whether they are causative mutations resulting in a loss of function or instead a gain of toxic properties. In addition, most ALS mouse models generated so far are transgenic made by randomly inserting human mutant ALS genes into the mouse genome. The expressions of these transgenes are well above endogenous levels and often lead to aberrant phenotypes in animal models[103]. Therefore, we generated both missense and frameshift deletion mutants. In this way, *PCDHA9* is expressed at physiological levels or is deleted, and these mutations should more closely recapitulate the human mutations at the genetic, molecular, and biochemical levels. Interestingly, both mutants and deletion homozygous mice showed pleomorphic phenotypes consisting of late disease onset, with progressive and asymmetric motor function deficits, including muscle atrophy and paralysis, which closely mimics the clinical manifestations of typical human ALS. An abnormal EMG is essential for the diagnosis of ALS in humans[58], and we detect it in all of our aged *PCDHA9* mutants. The significant loss of motor neurons in the spinal cord and segmental denervation of skeletal muscles are in accordance with the pathological hallmarks of ALS (Fig. 4g). Therefore, both clinical and pathological phenotypes detected in *Pcdha9* mutants make it an ALS-relevant animal model and demonstrate a potential connection of PCDHA9 functional domains with ALS.

Almost all *Pcdhα* family genes are expressed in neurons in the central nervous system. Studies in mice with null alleles or deletions in the conserved cytoplasmic region of the PCDH-α family demonstrated that this family of genes plays roles in the establishment of neural circuitry including axon outgrowth, targeting, and tiling, dendrite arborization, synaptogenesis, and neuronal survival in the CNS[39,41,104−106]. In addition, the clustered *Pcdh* genes have been shown to control neuronal migration, dendrite arborization, and cytoskeletal dynamics in forebrain neurons via signaling involving adhesion kinases, WAVE, FAK, PYK2, and RHO family GTPases[61,107−109]. Mice with deletion of the *Pcdhα* gene cluster show defects in cognitive and affective functions such as depression[110]. The function of individual genes has not been studied other than the double knockout of the two C-type genes or the deletion of whole family members[39,41,109,110]. However, the long-term effects of the loss of members of the *Pcdhα* family have not been reported in old animals. We found that both *Pcdha9* L700P and deletion mutants did not show significant motor dysfunction until 10-month-old but exhibited progressive motor function deficits after that. This indicates that PCDHA9 is essential for the survival of motor neurons and/or the maintenance of NMJs in old age. In addition, the activation of astrocytes in motor neuron-concentrated areas further supports the occurrence of motor neuron degeneration. As expected, the loss of motor neurons and/or denervation of NMJ led to the dramatic or severe wasting (atrophy) of skeletal muscles and subsequently the paralysis and death of old mutant mice (Fig. 4g). The late onset of phenotypes may be due to the compensated expression of other genes in the *Pcdhα* gene cluster (or *Pcdh-β, -γ* families) and activation of different signaling pathways in young mice. In support of this notion, triple deletion of the *Pcdh-α, - β*, and *-γ* families leads to the development of fewer motor neurons in the spinal cord in mice around birth[111].

Since both point mutation and deletion mice display similar ALS phenotypes, the L700P mutation is likely to be a loss of function that causes the loss of motor neurons. These mutations may also affect the function of other *Pcdh* cluster family proteins, considering the dimeric or multimeric properties of PCDH cluster proteins[112,113]. The damaging variants in one protein may interfere with the normal function of other PCDH cluster proteins and thus the survival of motor neurons.

L700P mutation of human PCDHA9 leads to its instability and dysfunction in suppressing the activation of FAK and PYK2. This would account for the hyper-phosphorylation of FAK and PYK2 in spinal cords and the loss of motor neurons in old Mut mice. In addition, we discovered the co-localization and interaction between PCDHA9 and NKA-α1. Consistent with a previous report (Edwards et al., 2013), NKA-α1 is highly expressed in the spinal motor neuron. Mutation of PCDHA9 results in dramatically reduced expression of NKA-α1 in CHAT-positive motor neurons in old mice. Since NKA-α1 is also highly expressed in the pre-synapse of NMJ in skeleton muscle[114], we can predict that the reduced function of NKA-α1 would contribute to motor neuron vulnerability and the dysfunction and/or loss of NMJs in our Mut mouse. Similarly, the impaired function of NKA-α3 presumably contributes to motor neuron vulnerability in SOD1 ALS mouse model and reduced expression was detected in the lumber spinal cord of ALS postmortem[115]. However, the expression levels of NKA-α3 and β1 are not affected in our old Mut mice. This suggests that NKA-α3 and β1 might compensate for the reduced function of PCDHA9 in young Mut mice.

To understand the underlying pathogenesis more comprehensively, we performed snRNA-seq and ATAC-seq on 9-month-old spinal cord before the onset of motor dysfunction. In the neuron cluster, where PCDHA9 is highly expressed, the GO analysis terms of snRNA-seq and snATAC-seq both indicate the dysregulation of synapse organization, ion transport and cell adhesion. In addition, the terms of regulation of neuronal death, oxidative stress response, and the association with neurodegeneration diseases in snATAC-seq analysis indicate that the process of cell death is about to take place or is taking place. Moreover, we mapped 49 genes encoding proteins that potentially interact with PCDHA9 and, meanwhile associated with significant DARs in snATAC-seq. These proteins are predominantly located on cell membrane or synapse and function as transporters. Some of other DEGs in RNA-seq dataset also have potential interactions with PCDHA9, such as *Ptprt* and *Plekhg1*. Of note, the interaction between FXYD2 and PCDHA9 in the mapping result and the protein-protein interaction between NKA-α1 and PCDHA9 indicate the disturbed function of NKA-α1 heterotrimeric complex in our Mut mice. Together, these data indicate that the signaling communication between cells or synapses was disturbed by the L700P mutation of PCDHA9 directly.

We then generated gene regulatory networks (GRNs) by linking distal cis-regulatory elements after integrating snRNA-seq and snATAC-seq datasets. We highlight the altered complexity of gene regulatory networks among different WT and Mut cell subtypes and important role of *Rxra* and *Malat1* in Mut neuron cluster. *Rxra*, a transcription factor, may induce the expression of *Malat1* which may control other genes' expression via its downstream miRNAs. Among the target genes of these miRNAs, we found five upregulated genes in our snRNA-seq dataset are associated with neurodegenerative diseases, including *Atcb, Elavl2*, and *Fgf1*, which are associated with ALS. In addition, analysis of the intercellular communication between cell types also provides a prediction on the affected receptor-ligand binding in our mouse model, which may contribute to the pathogenesis of ALS. Our bioinformatics analysis provides a perspective for understanding the mechanism underlying ALS, although further research is warranted to confirm those findings.

In summary, our orthologous point mutation mouse model based on the recurrent mutation found in ALS patients accurately recapitulated most neuropathological and clinical phenotypes of ALS. The late disease onset and progressive phenotypes make the *PCDHA9* mutant an ALS-relevant physiological system to explore disease mechanisms including the early onset of ALS, and to test potential therapeutics. Combined with multi-omics analysis, we identified the disturbed gene regulatory networks and signaling involved in cell/cell-cell adhesion, ion transport, synapse organization and neuronal survival in aged mutant mice.

## Limitations of the study

We tried to identify genetic contributors in a large cohort of ALS patients, the homozygous variant (p.L700P) in *PCDHA9* was found only in three unrelated patients. Although these patients were confined to Sichuan province in Southwest China, we have not performed the rare variant association testing that could both evaluate the null probability of observing the 3 homozygous carriers and cater for population structure. Moreover, the relatedness of the 3 patients was not assessed using the current panel sequencing data, since the number of variants included in these data was too small. For the same reason, there was not enough data to support a haplotype analysis to suggest different mutation events. Thus, based on the presented human genetic evidence, it is uncertain whether p.L700P is truly an incidental "Sichuan variant". Thus, more cases should be inspected in other regions around the world, and further validation of PCDHA9 as a disease-causing gene in ALS is needed.

## Methods
### Study participants
The participants in the discovery (whole-exome sequencing, 154 cases and 102 controls) study were recruited from West China Hospital, Sichuan University between May 2008 and August 2018. The subjects in the targeted sequencing study included those used in the discovery study and recruited from Xuanwu Hospital, Capital Medical University and The First Hospital Affiliated Hospital, Sun Yat-sen University (238 cases and 226 controls) between April 2012 to January 2019. The additional ALS cases (548 Northern and 375 Southern) tested in the Sanger sequencing were recruited from Xuanwu Hospital, Capital Medical University, the Chinese PLA General Hospital, and The First Hospital Affiliated Hospital, Sun Yat-sen University between March 2016 and November 2020. The demographic and clinical data were shown in Table S1. All participants were subjected to the clinical neurological assessments, and the patients were all evaluated by the needle EMG study. ALS was diagnosed according to the Gold Coast (GC) criteria for ALS[57] by at least two neurologists. The healthy controls did not have any nervous system or psychiatric diseases. Written informed consent was obtained from all the participants and blood samples were obtained afterwards. All investigations were conducted according to the Declaration of Helsinki, and the study was approved by the Ethics Committees of Xuanwu Hospital of Capital Medical University, West China Hospital of Sichuan University, The First Affiliated Hospital of Sun Yat-sen University and The Chinese PLA General Hospital.

### Whole-exome sequencing (WES)
WES was conducted on 500 ng of genomic DNA from cases and controls. Fragment libraries were created from the sheared samples by sonication and target enrichment was performed using Agilent SureSelect QXT ALL Human Exon V6 kit (64MB) (Agilent, Santa Clara, USA). Captured DNA was amplified followed by solid-phase bridge amplification and paired-end sequenced on Illumina Hiseq 2500 (Illumina, San Diego, USA). For targeted sequencing, we designed a gene panel incorporating frequently mutated genes identified in our exome sequencing, as well as genes reported to cause familial ALS and associated with sporadic ALS by association (candidate and genome-wide association studies) and next-generation of sequencing (whole-exome, whole-genome, and targeted sequencing) studies, according to the detailed information from the Amyotrophic Lateral Sclerosis Online Database-ALSoD (https://alsod.ac.uk/).

### Ultra-depth targeted gene sequencing by liquid-phase hybrid
A customized panel of 288 genes was selected including the identified genes with recurrent (≥2) mutations by our whole-exome sequencing (WES) study (110), the known ALS genes (25), genes identified by other WES (36), and association studies including GWASs (86), as well as genes functionally identified to contain RNA-recognition motif (RRM, 31) (Tab. S2).

The targeted region capturing and sequencing was performed as previously (Yu et al., 2017). In brief, using an automated SPRI works System (Beckman Coulter, San Jose, USA), the 200–500 bp size libraries were constructed with a TruSeq DNA Sample Preparation Kit (Illumina, San Diego, USA). The libraries were then submitted for capturing using TargetSeqTM liquid chip capture sequencing kits (iGeneTech, Beijing, China). After hybridization, the DNA: RNA hybrid was enriched by biotin-labeled magnetic beads and amplified in ABI 2720 (Thermo Fisher, Foster City, USA). The quality of the amplified library was determined using an Agilent 2100 Bioanalyzer (Agilent Technologies, Santa Clara, USA). Libraries with good quality and DNA concentration >3 ng/μl were sequenced using an Illumina Hiseq X-ten sequencer (Illumina, San Diego, USA).

## Bioinformatics analyses of the sequencing data

The adaptor sequences and raw reads with low-quality reads were filtered using Trimmomatic software[116]. The adaptor sequences were: 5′-GATCGGAAGAGCACACGTCT-3′; and 5′-AGATCGGAA-GAGCGTCGTGTAGGGAAAGAGTGT-3′. Low-quality sequences with a quality value < 20, accuracy of <99% or base length shorter than 40 bp were removed. Fastqc software (http://www.bioinformatics.bbsrc.ac.uk/projects/fastqc/) was used to measure data quality to ensure 95% of the remaining reads or clean reads with quality were more than Q30. The clean reads were then aligned to the reference human genome (hg19, GRCh37) using the Burrows-Wheeler Alignment Tool (BWA) 0.7.15[117] to generate BAM files. SAMtools and Picard software (http://broadinstitute.github.io/picard/) were used to remove PCR repetitive sequences and process these lane-level SAM files to generate a sample-level BAM file for variant calling. The Genome Analysis Toolkit (GATK) 3.8[118] was used to recalibrate base quality scores and detect the variants such as SNPs and InDels in BAM files. Finally, ANNOVAR software (released on 2020-6-8)[119] was used to annotate the variants. Novel variants were filtered against 1000 Genomes (1000 genomes release phase 3, (http://www.1000genomes.org/), dbSNP (https://www.ncbi.nlm.nih.gov/projects/SNP/snp_summary.cgi), Exome Aggregation Consortium (ExAC) (http://exac.broadinstitute.org), and Genome Aggregation Database (http://gnomad.broadinstitute.org). Rare damaging variants were defined using the following criteria: (i) call rate ≥ 95%; (ii) allele minor allele frequency (MAF) in the general populations in ExAC and gnomeAD ≤ 0.1%; (iii) in silico prediction to be damaging or deleterious using Polyphen 2 software (http://genetics.bwh.harvard.edu/pph2/) and SIFT program (http://siftdna.org/); (iv) according to the guidelines from the American College of Medical Genetics and Genomics (ACMG) and the Association for Molecular Pathology (AMP), and criteria dynamically updated by the ClinGen Sequence Variant Interpretation (SVI) Workgroup. We used REVEL (rare exome variant ensemble learner) for predicting the pathogenicity of missense variants, and multiple online tools (nonsense-mediated mRNA decay (NMD), SpliceAI, etc) for predicting the loss-of-function (LoF) changes in genes (PVS1 evidence) for null variants (stop gain, splicing donor and frameshift).

## Sanger sequencing for validation of the L700P mutation

To validate the *PCDHA9* L700P mutation in additional ALS cases, we used the following primers: forward: 5′-CTTTCATACGAGCTGCAGCCA-3′; reverse: 5′-ATGAGGTCG GTCTTCTGCTTA-3′.

## Generation of *Pcdhα9* point mutation and deletion mice

This model was generated by Beijing Biocytogen Co., ltd. All mice were in C57BL/6 background, except for the mice in Fig. 7c (The mice in Fig. 7c were crossed with HB9-GFP transgenic mice). Mice were housed in a 12 h light/12 h dark cycle. Housing temperature ranged from 21 to 23 °C. Housing humidity ranged from 30 to 70%. The animals used, both knock-in and deletion mutants, were descendants of heterozygotes mating with heterozygotes. For the genotype of mice, we have sequenced the mouse DNA to verify the successful establishment of both Mut and Del mouse models using CRIPR/CAS9 (Fig. S1c). They were kept separately to avoid mixed breeding. Genotype was inspected again for each descendant of those heterozygotes. Tail biopsies were used to extract genomic DNA for genotyping. The primers used for cloning genes including:

WT hPCDHA: 5′-CCGCTCGAGATGTTATACTCAAGTCGAGG-3′ & 5′-CTAGTCTAGATCACTGGTCACTGTTG-3′.

Point mutation Mut hPCDHA: 5′- GATGTCAACGTGTACCCGAT-CATCGCCATCTGC-3′ & 5′-GCAGATGGCGATGATCGGGTACACGTTGA CATC-3′.

When the genomic DNA only showed WT hPCDHA9 band was WT mouse, both showed WT and Mut bands was heterozygous mouse, and only showed Mut hPCDHA9 band was homozygous Mut mouse.

All experimental procedures were performed according to protocols approved by the Institutional Animal Care and Use Committee at the Institute of Genetics and Developmental Biology, Chinese Academy of Sciences. Mice were sacrificed when they were unable to move or had breathing problems.

## Quantitative real-time PCR

RNA was extracted by Trizol, 1.5 μg RNA was employed for cDNA reverse transcription, and SYBR Green mixture was used for the reaction system. The specificity was determined through the melt curve. All data were normalized by GAPDH transcript and calculated using the ΔΔCq method. The primers used for mouse *Pcdhα9*:

5′-AGTCTTACACCTTGCCCAGTGG-3′ & 5′-TGTGCATGCCTGCTC TTAGCGA-3′; human *Pcdhα9*:

5′- CTGCCACATCTTCACGGTGTCT-3′ & 5′-TGCACTGACACGTAG CTCGACA-3′; mouse GAPDH:

5′-CATCACTGCCACCCAGAAGACTG-3′ & 5′-ATGCCAGTGAGCTT CCCGTTCAG-3′.

## Behavioral tests

Behavioral tests were chosen and performed according to Brooks' review[120]. No sex- and gender-based analyses have been performed in this study, because there were no significant gender differences in the ALS diseases involved in this study, we did not think that gender would cause phenotype differences in mouse experiments. Both male and female mice were used for survival rate (in SPF room), while only unmated male mice were used for the behavior tests to avoid the impact of the physiological cycle of female mice. The animal behavior analysis was performed in the clean grade room. All the tests were performed between 8:00–11:00 am.

**Footprint.** The test is used to analyze the gait of mice. The flat passageway used was 70 cm long and 7 cm wide. Red and blue pigments were painted on mice's forelimbs and hind limbs respectively. Measuring the step stances including stride length of the left hind limb, left forelimb, right hind limb and right forelimb, forelimb stance, and hind limb stance. Every index needs an average of more than 6 steps for the statistical analysis.

**Grip force.** The grip force test is used to detect the grip force of the forelimbs or all limbs. The grip force was measured using a digital grip strength meter (Mark-10 Corporation, Copiague, NY)[121]. A fine metal grid was fixed on the sensor. The peak tension value was recorded. In the dragging process, the mouse body was kept parallel with the grid. The value for each animal is an average from 3 times of tests, with rest for 5 min between each test.

**Swimming test.** This test is used to assess limb coordination in voluntary exercise. The transparent sink used was 1 meter long the full of water at room temperature with a high platform at the end of the tank for mice to rest. A 70-cm long distance from the other end to the platform was used to record the time and used for the statistical analysis.

**Rota rod test.** This is the test for motor coordination and balance of mice and is also fit for lateralized impairments. The minimum speed is 5 rpm and the maximum speed is 50 rpm, which is accelerated uniformly within 5 min. Before starting, the mice need posture adjustment for 30 sec at 5 rpm. Recording the time when the mouse drops. The continuous rotation over more than two cycles is also regarded as falling off. If the mouse falls off in the beginning 50 sec, a second chance will be given. Every mouse was tested 3 times/day, a break of at least five minutes between sessions will be given. The average value is used for statistical analysis.

## Cells culture and plasmid generation

HEK 293 cell line culture: Cells were cultured by 10% FBS in DMEM. Transfected reagent using VigoFect (Vigorous, Beijing). The culture medium is mixed with CHX at the concentration of 100 μg/mL or MG132 when needed. CHX (C112766, Aladin, 0.1 mg/mL); MG132 (1748, Tocris bioscience, 0.1 mg/mL). The hPCDHA9 (gifts from Jiahuai Han's lab) that fused with the Flag tag at N-terminal and inserted into the pCMS-EGFP vector. Pyk2 and Fak were gifts from Wu lab in Systems Biomedicine (Ministry of Education), Shanghai Jiao Tong University. All plasmids were validated by sequencing and restriction enzyme analysis. For the point mutation human PCDHA9 generation, below primers were used: 5'-GATGTCAACGTGTACCCGATCATCGCCATCTGC-3' & 5'-GCAGATGGCGATGATCGGGTACACGTTGACATC-3'. The point mutation was validated by sequencing.

## Immunohistochemistry and immunofluorescence analysis

Mice were anesthetized by tribromoethanol and perfused with a phosphate buffer solution (PBS) followed by 4% paraformaldehyde (PFA). The lumbar 3–5 spinal cord was collected and fixed in 4% PFA overnight. The gastrocnemius muscles were collected and weighed, then fixed in PFA following dehydrating by graded ethanol.

For histological analysis, paraffin sections were stained by hematoxylin for 8 min and eosin for 1 min. After sealing, sections were observed, and images were taken with Nikon TE2000-S.

For immunofluorescence analysis, frozen spinal cord sections were permeabilized with Triton X-100 (0.1% in PBS) following blocking buffer (10% FBS and 5% BSA in PBS), incubated with primary antibodies overnight at 4 °C. Then incubated with secondary antibody conjunction with Alexa fluor®. Antibodies used are as followed: CHAT (ab178850, Abcam. GR3230471-1. Clonal number: EPR16590. 1:1000), GFAP (Dako, Z0334, 20044021. 1:1000), NeuN (Abcam, ab104224, 1030423-1. Clonal number: 1B7. 1:1000), Na$^+$/K$^+$ATPase-α1 (ab76020, Abcam, 1:1000), NKA-α1 (ab7671, Abcam, 3294995-5, Clonal number: 464.6; 1:1000), TDP43 (Abcam, ab237270, GR3254079-1. Clonal number: EPR18554). For immunostaining of the transmembrane signals in cultured 293 cells: cells transfected (for 18 h) were changed the culture medium with the PCDHA9 (Proteintech, 18075-1-AP, 01002. 1:300) primary antibody solution and incubated for 30 min at 37 °C. Then fixed by 4% PFA following routine staining procedure. Images were photographed by LSM 700 (Carl Zeiss) confocal microscope and analyzed by ImageJ (1.8.0).

## Western blot and immunoprecipitation

The procedures of Western blot and immunoprecipitation were the same as the described previously[122]. Concisely, tissues or cells were collected and lysed by lysis buffer (50 mM Tris-HCl, 150 mM NaCl, 1% NP40, 0.5% DOC, 1 mM EDTA) supled with 2 mM PMSF, protein inhibitors and phosphatases inhibitors cocktails. Then samples will be collected by centrifuge with 13,000 g for 15 min at 4 °C after 15 min lysed on ice. For western blot, the supernatants were boiled at 95 °C for 5 min in SDS protein loading buffers directly. For IP, the supernatants were incubated with Flag beads overnight at 4 °C with rotation, and then the supernatants were collected by centrifugation at 2300 rpm for 2 min at 4 °C. Beads were washed three times then boiled at 95 °C for 5 min in SDS protein loading buffers and analyzed by western blot.

Equal amounts of protein were separated by on 10 or 12% gradient SDS-PAGE gels and transferred to nitrocellulose membranes. Membranes were blocked with 5% skim milk for 1 h at room temperature and then incubated with the relevant primary antibodies overnight at 4 °C. After washing, the membranes were incubated with HRP-conjugated secondary antibodies in 5% skim milk for 1 h at room temperature. Membranes were washed and signals were developed using ECL Plus western blot substrate. Western blotting images were obtained using SageCapture software and analyzed using Lane 1D for the gray value.

The antibodies used are as follows: P-FAK (3283, CST, 1:1000); His (D291-3, MBL, lot: 008; 1:2000); P-PYK2 (CST, 3291 S, lot: 5, 1:1000); Myc (CST, 2278, lot: 7, 1:1000); P-FAK (CST, 3283, lot: 6, 1:2000); Flag (M185, MBL, lot: 001; 1:2000); GFP (ab13970, Abcam, GR3361051-15, 1:1000); NKA-α1 (ab7671, Abcam, 3294995-5, Clonal number: 464.6; 1:1000); NKA-α3 (ab182571, Abcam, Clonal number: EPR14138. 1:1000); NKA-β1 (ab76020, Abcam, 1:1000); α-TUBULIN (CST, 3873, lot: 6, 1:5000). PCDHA9 (Proteintech, 18075-1-AP, 01002. 1:1000; Polyclonal).

## Electromyogram (EMG) acquisition and analysis in anesthetized mice

Mice were anesthetized by intraperitoneally injecting 10% of chloral hydrate (0.3 mg/kg) and placed supine or prone on a heating pad maintaining the temperature at 37 °C. The method to assess the depth of anesthesia is to use tweezers to clamp the foot's pad to observe withdrawal response. Concentric needle electrodes (30 g*25 mm, Natus, USA) were inserted into each of the four quadrants of the examined muscle in turn to check EMG activity. After removing hair from the hind limb, subdermal electrodes (10 mm × 0.30 mm, Alpine BioMed, USA) connected to an amplifier were inserted into the area of the lower extremity muscles near the sciatic nerve to record compound muscle action potential (CMAP) when using fine ring electrodes placed at the distal end of the lower extremity for stimulation. The recording electrode placed at the distal end was 1 cm away from the proximal reference electrode. Electrical stimulation was conducted with a 0.1 ms bi-directional plan. Sensory nerve action potential (Bregin et al.) was recorded by ring electrodes binding on lower limbs distal when stimulated by subdermal electrodes. A disposable surface electrode was placed on the tail for grounding. All recordings were made using a Medtronic key point EMG machine (Keypoint.net system v 2.21, DK).

## Isolation and characterization of NMJs

The NMJs were isolated and characterized according to the protocol as reported[121]. Fresh semitendinosus muscle was taken after mouse anesthesia. Then the entire muscles were incubated with Alexa Fluor 594 conjugated α-bungarotoxin (α-BTX-Alexa 594) (Thermo Fisher Scientific, B13423, 1834760, 5 μg/ml) for 20 min and post-fixed in 4% paraformaldehyde (PFA) overnight. The whole mount muscle was stained the pre-synaptic axon by anti-neurofilament 200 (Sigma, N4142, 088M4801V, 1:400) for 48 h, followed by incubation with a second antibody for 4 h. Stained muscles were embedded in Tissue-Tek O. C. T. Compound, and longitudinal sections of 50 μm thick were cut. Images were taken by ZEN LSM700 confocal.

## Toluidine blue staining

The procedures of TEM slice were described as previously[123,124]. Concisely, isolated sciatic nerves were fixed with 2.5% glutaraldehyde, post-fixed in 1% osmium tetroxide, dehydrated in acetone, and then embedded in EPON812 resin. Sliced into 500 nm, 37 °C dried. Toluidine blue stained for 1 min at 37 °C. Photos were taken with Nikon TE2000-S. The diameter of axons in sciatic nerves was measured by ImageJ software.

## RNAscope technology

RNAscope® technology is an innovative multichannel second-generation fluorescence detection method for in situ hybridization. Protocols followed the instructions[125]. Mouse was perfused by PFA after being anesthetized, and L3-L5 spinal cord was collected and dehydrated by 30% sucrose. After being embedded by OCT, the spinal cord was sliced into 20 μm sections and pasted on the adhesive slide directly. Then incubated with hydrogen peroxide (3% $H_2O_2$ in methanol), 100 °C retrieval solution, and 100% ethanol successively. After that, treated with RNAscope® protease III and incubated with pre-heated probes (C1:C2 = 50:1). C1 is the CHAT probe (408731, ACD) and

C2 (540581-C2, ACD) is the *PCDHA9* probe. Then incubated with AMP1, AMP2, AMP3, HRP-C1, TSA®plus cy3, HRP stocking, HRP-C2, TSA®plus cy5, HRP stocking and DAPI according to the manufacturer's instructions. Images were taken by ZEN LSM700 confocal microscope.

## Nuclei isolation from frozen spinal cord tissue

The frozen ventral horn of the L3–L5 spinal cord was used for the isolation of the nuclei. The spinal cord was taken from the 9-month-old male mice and quickly placed straightly in the embedding mold. Then the spinal cord was quickly frozen with OCT in liquid nitrogen. Removed the ventral white matter in the microtome until the gray matter appears and removed the white matter on both sides carefully. Then taking the gray matter at the depth of 1/2 before the confluence of the gray matter of two sides and put the gray matter into the EP tube for liquid nitrogen quick freezing again. Then the nuclei were extracted. Ground the tissues in 500 μL homogenization buffer [0.32 M Sucrose solution, 0.01% NP40, 0.1 mM PMSF (Roche), 1 mM β-ME-2-Mercaptoethanol] and centrifuged in 300 g for 5 min at 4 °C, removed the upper turbid liquid and washed sediment twice with 1 mL HB. Resuspended the sediment by 300 μL HB and added isochoric 50% Medium, pipetted to mix sufficiently. Then added 700 μL 29% Medium to the bottom, and centrifuged in 3000 g for 10 min at 4 °C with a swinging bucket centrifuge to obtain the bottom sediment nuclei. 0.32 M Sucrose solution: 10 mM Tris pH 8.0, 320 mM sucrose, 5 mM $CaCl_2$, 3 mM Mg (Ac)$_2$, 0.1 mM EDTA, 1% nuclease-free BSA, and 1:200 RNAase inhibitor (RI) (Takara, 2313 A). 50% Medium (900 μL): 750 μL 60% OptiPrep (Sigma, D1556), 150 μL RNA-free water, 9 μL RI. 29% Medium (800 μL): 400 μL 58% OptiPrep, 400 μL 0.32 M sucrose solution, 2 μL RI. 58% Medium (600 μL): 580 μL 60% OptiPrep, 20 μL BSA, 6 μL RI.

## Tandem mass tag (TMT)- labeled quantitative proteomic analysis

Mice were sacrificed after anesthesia by isoflurane following neck cutting. The spinal cord L3–L5 was taken and frozen in liquid nitrogen immediately and the ventral horn of the bilateral spinal cord (about 400 um thick in gray matter area) was collected at low temperature. The samples were digested with Ripa lysate (containing 4% sodium dodecyl sulfate, protease inhibitors and phosphatase inhibitors). The protein concentrations were detected using a BCA protein assay kit (Thermo Scientific, Rockford, IL). Protein digestion was performed using the filter-aided proteome preparation (FASP) method. After digested with sequencing grade trypsin (Promega, Madison, WI) at 37 °C overnight, the resultant tryptic peptides were labeled with acetonitrile-dissolved TMT reagents (Thermo Scientific, Rockford, IL) by incubation at room temperature in dark for 2 h. The labeling reaction was stopped by 5% hydroxylamine, then equal amounts of labeled samples were mixed before prefractionation with reversed phase (RP)-high performance liquid chromatography (HPLC). Sample prefractionation was performed using an offline basic RP-HPLC approach. The LC-MS/MS analysis was performed using an Orbitrap Fusion™ Lumos™ Tribrid™ mass spectrometer (Thermo Scientific, Rockford, IL Waltham, MA) coupled online to an Easy-nLC 1200 in the data-dependent mode. The database search was performed for all raw MS files using the software MaxQuant (version 1.6). The Mus musculus proteome sequence database downloaded from uniProt (https://www.uniprot.org/) was applied to search the data. The functional results were analyzed by GO analysis.

## Single nuclei RNA-seq and single nuclei ATAC-seq library preparation and sequencing

Each 10,000 nuclei were used for the snRNA or snATAC library construction.

For snRNA-seq, the single cell 3' GEM, Library & Gel Bead Kit V3.1 (10× Genomics, 1000075) and Chromium Single Cell B Chip Kit (10×

Genomics, 1000074) were used. To generate single-nuclei gel beads in emulsion, the nuclei suspension was loaded onto the Chromium single cell controller (10× Genomics). Then suspended the single nuclei in PBS (containing 0.04% BSA). Captured cells were lysed to release their RNA and barcoded through reverse transcription in individual GEMs. The reverse transcription was performed on a S1000TM Touch Thermal Cycler (Bio-Rad) at 53 °C for 45 min, followed by 85 °C for 5 min. The cDNA was kept at 4 °C and then amplified for sequencing.

For snATAC-seq, incubating the nuclei with Tn5 transposase. Then the nuclei suspension was loaded into the Chromium microfluidic chip E with 10x Genomics reagents and barcoded with a 10x Genomics Chromium Controller (10x Genomics, Pleasanton, CA). DNA fragments were subsequently amplified, and the sequencing libraries were constructed with reagents from a Chromium Single Cell ATAC reagent kit (10x Genomics; PN-1000110, PN-1000156, PN-1000084) according to the manufacturer's instructions. After preliminary quantification and quality inspection, libraries were then pooled and loaded on an Illumina NovaSeq with 2 × 50 paired-end kits.

## Data processing of snRNA-seq

The single-nucleus RNA sequencing (snRNA-seq) data was processed using Cellranger (v5.0.1) and mapped to the 10X reference for mm10 (v1.2.0). We removed cells with an estimated percentage of contamination greater than 70% calculated by Celda (v1.9.3)[126] using the DecontX function. The estimated percentage of contamination was visualized using UMAP plots. We then filtered out low-quality cells and potential doublets using Seurat (v4.1.1), which removed cells that expressed less than 500 genes, more than 60,000 UMIs, and had mitochondrial gene percentages over 5%. We also excluded cells that expressed hemoglobin genes (Hbb-bs, Hbb-bt, Hbbt1, Hbbt2, Hbb-y, Hba-x, Hba-a2, Hba-a1, Hbq1b, Hbq1a). After basic quality control, the data was normalized and scaled using the SCTransform (flavor=v2) function. We then used the FindClusters and FindNeighbors functions with a resolution of 0.5 to cluster cells based on strong PC1–PC15.

## Integrate human snRNA-seq by seurat3.0

To further analyze the snRNA-seq data of WT and Mut mouse, we integrated these two data by seurat3.0[127]. For integration, 3000 shared highly variable genes were identified using Seurat's 'SelectIntegrationFeatures function. Integration anchors were identified based on these genes using preSCTIntergration and FindIntegrationAnchors function. PCA was again carried out, and the top 15 PCs were retained. The clustering was again performed with the clustering resolution 0.5.

## Identification of cell type in snRNA-seq and identification of DEGs among samples

The marker genes of each cluster were identified by using the FindAllMarkers function (thresh.use = 0.25, only.pos = TRUE, min.pct = 0.25, logfc.threshold = 0.5, test.use = "wilcox"). And we used multiple well-known cell-type-specific marker genes to identify each cell type, Cx3cr1, and Trem2 for the microglia; Mbp and Mobp for the oligodendrocyte; Slc1a2 and Slc1c1 for the astrocyte; Pdgfra and Vcan for the OPC and Rbfox1 and Snap25 for the neuron.

To get DEGs among samples, we use the PrepSCTFindMarkers function which ensures that the fixed value obtained by previous steps is set properly. Then we use FindMarkers (assay = "SCT", logfc.threshold = 0.2) to find differentially expressed genes, and only genes that highly expressed in well-conditioned cells were considered.

## GO enrichment analysis

We used clusterProfiler (v4.1.4)[128] to get enriched GO terms of the genes we focused on. We picked genes ($P < 0.05$) found in previous studies and used enrichGO function (ont = "BP", pAjustMethod = "BH", pvalueCutoff = 0.05, qvalueCutoff = 0.05) to perform GO.

## Cell-cell communication analysis

Cell-cell interactions between each cluster were identified by Cell-PhoneDB (v3.0)[90]. We perform lift-over between the mouse and human genome by downloading the orthologous gene list from Ensembl V104 (http://asia.ensembl.org/biomart/martview/7d9c5326895fb3214edd7796dcae1bc8) and the significant cell-cell interactions were selected with $p$-value < 0.05. The mean value in Fig. 8h is calculated by default parameters and mean value in Fig. S9h is calculated by log2 (mean +1).

## snATAC-seq analysis

Raw sequencing data were converted into the FASTQ format using the cellranger-atac pipeline (v.1.2.0). The Mm10 reference genome was used for data alignment and used for generating the FASTQs. For mapping and chromatin accessibility, the cellranger-atac count command was used. Fragment data were further processed using ArchR (V2.0.1)[129]. We remained the cells with transcription start site (TSS) enrichment scores more than 4 and fragment numbers more than 10^3.5. Doublet analysis was performed by using addDoubletScores and filterDousblets functions in AchrR. According to the gene scores of well-known marker genes and predictions of gene expression based on the accessibility of regulatory elements calculated by AchrR, we defined 5 different cell types, Cx3cr1 and Trem2 for the microglia; Mbp and Mobp for the oligodendrocyte; Slc1a2 and Slc1c1 for the astrocyte; Pdgfra and Vcan for the OPC; Rbfox1 and Snap25 for the neuron. Then we performed Seurat3.0[127] to integrate Mut and WT snATAC-seq data. "Iterative LSI" and major pc2-pc15 were used to cluster by Seurat's FindClusters function with resolution 0.2. We identified different cell types by calculating gene activity scores for several well-known reported marker genes. 148662 peaks were found using MACS2 (v2.1.1)[130]. GetMarkerFeatures function (useMatrix = "GeneScoreMatrix", bias = c ("TSSEnrichment", "log10 (nFrags)", testMethod = "wilcoxon")) and FDR < = 0.01 & |$Log_2$FC| >=1.25 were performed to find significant DARs among clusters in snATAC-seq. GetMarkerFeatures function (useMatrix = "PeakMatrix", bias = c ("TSSEnrichment", "log10 (nFrags)", testMethod = "wilcoxon", testMethod = "binomal")) and FDR < = 0.05 & |$Log_2$FC| >=1 were performed to find significant DARs among samples in snATAC-seq. The genes associated with target genes were predicted by geneMANIA (http://genemania.org/) and protein–protein interaction[131] was analyze by STRING (string-db.org). In Fig. 8d, GO analysis was performed using BiNGO[132] in Cytoscape (v3.9.0). $p$ values were adjusted by Benjamini-Hochberg FDR correction and FDR < 0.05 cut-off was used to determine significant enrichments. And networks were visualized in Cytoscape. In Figs. S9d, e, we used FDR < = 0.01 & |$Log_2$FC| >=1 parameter to get DARs and we used clusterProfiler (v4.1.4) to perform go analysis.

## Construction of gene regulatory networks

The construction and analysis of gene regulatory networks in our study involved a three-step process: (1) the examination of snATAC-seq and snRNA-seq datasets; (2) the integration of these datasets; and (3) the inference of cis-regulatory interactions to define a transcription factor-gene regulatory network (TF-gene GRNs).

In terms of single-cell transcriptomes and epigenomes data integration, we employed the "addGeneIntegrationMatrix" function to incorporate the gene expression matrix of snRNA-seq data onto the "geneScoreMatrix" of snATAC-seq data in both WT and Mut mice.

The method utilized for the subsequent definition of TF-gene GRN, involving the inference of cis-regulatory interactions, is complex and a separate manuscript is currently being prepared in the laboratory of Dr. Qiang Tu. The insights gained from GRNs of our study serve as preliminary findings and may provide potential directions for future hypothesis generation rather than as definitive conclusions.

## Statistics and reproducibility

The study was first designed to screen rare mutations in sporadic Chinese ALS patients by whole-exome sequencing (WES) and validated by targeted gene sequencing. Based on the bioinformatic analyses, the recurrent homozygous rare coding variants were picked out and functionally tested by knock-in mouse models. In human studies, no statistical method was used to predetermine sample size, and randomization and blinding was not used. Sex and/or gender was not considered in the study design and the information on sex and/or gender of participants was determined based on self-report. Samples with low quality DNA samples and sequencing data were excluded from the analyses.

For the animal, tissues and cells experiments, all statistical analyses were performed using Prism 9.0 or 10.1.0 (GraphPad Software, Inc). Data were presented as mean ± S.E.M. Sample sizes were selected to ensure sufficient statistical power while minimizing the number of animals used, randomization and blinding was used. No statistical method was used to predetermine sample size. Animals that were successfully measured were not excluded from the analysis. No data were excluded from the analyses. The $t$-test was used to compare two groups, and one way ANOVA was used for comparisons involving more than two groups. The significance level was set at $P < 0.05$. Detailed statistical information is provided in the figure legends.

## Reporting summary

Further information on research design is available in the Nature Portfolio Reporting Summary linked to this article.

## Data availability

The raw sequence data from both snRNA-seq and snATAC-seq have been deposited in Genome Sequence Archive[133] in National Genomics Data Center[134]. The human whole-exome sequencing and targeted gene sequencing data have been deposited in the China National Center for Bioinformation / Beijing Institute of Genomics, Chinese Academy of Sciences (GSA: CRA007917; HRA003114) that are publicly accessible at https://ngdc.cncb.ac.cn/gsa. Additionally, the raw sequence data and the processed data derived from snRNA-seq and snATAC-seq are conveniently accessible through the Gene Expression Omnibus under the SuperSeries accession GEO: GSE234783. Furthermore, the mass spectrometry proteomics data have been deposited to the ProteomeXchange Consortium through the iProX partner repository with the dataset identifier PXD042929. Source data are provided with this paper.

## Code availability

The code utilized for our analysis is available in https://github.com/NeuroXplorer-XuLab/PCDHA9-ALS-Candidate-Gene-Functional-Identification[135]. All code and analysis are available on Zenodo. The DOI is https://doi.org/10.5281/zenodo.10561493.

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

## Acknowledgements

We would like to thank Drs. A. Gitler, Y. Jia and Q. Wu for their very helpful advice. This study was supported by grants from National Natural Science Foundation of China (NSFC) (32061143026, 31921002,

32330038, 32394030), Instrument R&D project of the Chinese Academy of Sciences (YJKYYQ20200052), and Major Projects of the Ministry of Science and Technology (2021ZD0202300) to Z.X.; NSFC (82171412) to C.W.

## Author contributions
Z.X., C.W., and Y.X. jointly directed the research, design the experiments and write the manuscript. J.Z., D.Z., Y.W., F.L., C.X. and S.K.A. designed and performed most of the experiments. D.Z., J.Z. and S.Z. performed all the bioinformatic analyses. X.Y., Q.Z., X.H., R.H., X.L., Q.Li, M.W., W.Z., Z.W. collected and diagnosed all the patients and involved in the analysis of genetics data. Everyone helped in the revision of the manuscript.

## Competing interests
The authors declare no competing interests.
