## [Peer Review File · Nature Communications]

PCDHA9 as a candidate gene for amyotrophic lateral sclerosisEditorial Note: This manuscript has been previously reviewed at another journal that is not operating a transparent peer review scheme. This document only contains reviewer comments and rebuttal letters for versions considered at *Nature Communications*. Mentions of the other journal have been redacted.

REVIEWER COMMENTS

Reviewer #1 (Remarks to the Author):

As one of the reviewers of the previous version of the manuscript I'm pleased to note that the authors have been very responsive to the comments raised and have significantly improved their manuscript with their revisions. There are no outstanding issues that I have concerns over and congratulate the authors for their execution of this excellent manuscript.

Reviewer #3 (Remarks to the Author):

I maintain my position from previous review that the authors have conducted a very extensive study and identified interesting results that will be of notable interest. Various limitations were highlighted in the previous review that warranted clarification. The authors have made numerous additions and provided a very detailed response. My comments regarding these are detailed below.

The authors response to my comments as reviewer 3

- The human genetics evidence was not strengthened but the limitations of the study concerning the potential incidental nature of p.L700P are now very clearly acknowledged. I had not intended to indicate that statements such as “we have not performed rare variant association testing”, “the relatedness of the 3 patients was not assessed using available genotype data” and there was “no haplotype analysis” should be added to the text. If the authors have genotype data for the cohort that they report on and access to a bioinformatician with experience in population genetics then these tasks could likely be performed within the span of an afternoon. It is also now inconsistent that the authors add a line to their limitations section stating “the relatedness of the 3 patients was not assessed” while still referring to “three unrelated cases” throughout the manuscript. My assumption is that the authors indicate that there is “no known relationship between the 3 carriers”. However, the authors should confirm that these samples do not exhibit easily detected immediate familial relationships using KING (<https://www.kingrelatedness.com/>), plink (<https://www.cog-genomics.org/plink/1.9/ibd>) or an equivalent software package and the exome sequencing data. If it is not straight forward for the authors to conduct a rare variant association test then I would propose that they could also instead evaluate if the PCDHA9 locus is supported by homozygosity mapping (<https://www.homozygositymapper.org/>, again this can be done extremely quickly and could add more weight to the nomination of L700P). In my previous review I indicated that if the human genetics evidence could not be strengthened then the limitations should be stated more clearly, however quick checks such as these should be performed unless there is a specific reason that it is not possible.
- The observation of TDP-43 pathology is an extremely strong and interesting addition to the manuscript. The images look very compelling, but a formal quantification is lacking. Could the authors

provide some kind of simple statistical comparison? Eg number of nuclei exhibiting TDP-43 depletion in mutant vs wildtype.

- I made some observations about inconsistencies in phenotypic severity concerning data in Fig2B and statements in the main text. The authors provide some explanation for this which concerns heterogeneity in the quality of housing conditions for animals assigned to different experiments. This is an important point as the severity of phenotypes is very relevant to the reader's interpretation. The authors add one line to the methods but a more detailed statement should be added to the main text to clarify these discrepancies for the reader.
- I commented that methodological details describing the building of gene regulatory networks were not provided. The authors indicate that the approach is "very complicated" and "in preparation for a ms". As a reviewer I can therefore offer no opinion as to whether the methodological approach for this section appears appropriate. However, as indicated in my previous review I also do not see that these sections of the manuscript are that critical to the overall story. They are interesting and relevant but do not yield insights that are substantiated and arguably receive too much attention. I think it is fine to include these analyses as tentative hypothesis generating analyses, so long as they are clearly indicated as such.

The authors response to issues raised by reviewer 2

- Clarified details concerning the breeding strategy should be added to the methods
- The rationale for the deletion line that was added on lines 215-217 is not entirely clear. Why is the fact that most of the variants identified in ExAC/gnomad are heterozygous relevant? Is this being interpreted as an indication of relevant selective constraint? If so gnomad generates predicted constraint metrics for genes in a more formal manner (pLI scores, available on their website), however the scores for PCDHA9 suggest reasonably high tolerance for loss of function of variants with multiple premature termination / frameshift variants occurring throughout the gene (including several observation of homozygous genotypes). In any case, as the authors suggest it seems relevant to contrast L729P (which may exhibit loss or gain of function) with a deletion variant for interpretation purposes. I feel this is the rationale that is required on lines 215-217 so that the reader is not given the impression that the deletion is being immediately treated as ALS relevant a priori.
- In my opinion a detailed longitudinal account of disease progression in the mouse models over time is not entirely clear. Reviewer 2 focused on this at length. Many comments were addressed, but interpretation remains complicated by the fact that mice kept in the clean grade animal room exhibited poorer health than mice used for survival analyses. This disconnect between survival analyses and other phenotypic characterizations needs to be explained in the main text as otherwise the reader is guided to incorrect conclusions about the time course of events. However I think the overall message that mice developed a robust late onset phenotype with evidence for progression and multiple key ALS associated hallmarks is already a clear compelling message that warrants reporting.

Reviewer #4 (Remarks to the Author):

The clarifications provided by the authors help in the understanding of the study itself and help to

address some of the limitations raised by the reviewers.

Overall, the homozygous and deletion mutants are helpful in demonstrating a mouse phenotype and the behavioral and phenotypic analyses are quite thorough for the symptomatic animals.

Major Concerns:

1. Given that PCDH gene clusters have been implicated in neurodevelopmental disorders (as the authors note in lines 112-116, additional pathological or electrophysiological analyses beyond the motor neuron counts performed at age 10 months would have provided additional insights into whether mutations or deletions in this gene were relevant in NMJ maintenance. Indeed, data in other models (including the SOD1 mouse model) suggest that there is a dying back phenomenon that predates motor neuron loss and phenotypic changes.
2. An important limitation to the study is the authors note in the response to reviewers that mice used in behavioral studies were kept in a clean grade room but those in the survival analysis were kept in the SFP room. This is an important finding that should be delineated in the “limitations of the study” section. This is noteworthy as other ALS models have also noted differences in phenotypes depending on the geographical and/or cleanliness of the animal rooms.
3. The TDP immunostaining in figure 4 is unconvincing. The significant cytoplasmic staining (and aggregation) of TDP43 in the WT animals would be very atypical and it is not clear that it is particularly more abundant in the mutant animals. If the authors choose to speculate on TDP43 pathology/biology, there are a number of strategies that should be employed.
4. The authors note in their response that mice that developed motor deficits all died. However, it is unclear why a large percentage of mice were unaffected either phenotypically or by survival. The responses of the authors are mostly restatements of the methods but don't offer insights into why such a large percentage of mice were unaffected and whether there are pathological correlates.

Minor concerns:

1. In lines 222-223, the authors may speculate on why paralysis started unilaterally in these mice.
2. The ChAT immunostaining in 4a and 4c look very different with a more diffuse staining in 4A (which I would expect) and a much more punctate staining in figure 4C (one wonders about autofluorescence).
3. In lines 237-241, the authors speculate that differences in the swimming and rotarod test when compared to grip strength may have to do with aging. An alternative hypothesis may be that other non-motor pathways that correlate with balance may be affected and subsequently manifested in rotarod times.
4. In lines 270-273 the authors argue that the finding of PCDHA9 by in situ hybridization is consistent with its important function in the survival of motor neurons. These data only suggest that it is expressed in motor neurons and does not implicate any importance in MN survival.
5. In lines 313-316, there is no quantification of WB to show that protein levels are “significantly” lower than that of WT.

REVIEWER COMMENTS

Reviewer #1 (Remarks to the Author):

As one of the reviewers of the previous version of the manuscript, I'm pleased to note that the authors have been very responsive to the comments raised and have significantly improved their manuscript with their revisions. There are no outstanding issues that I have concerns over and congratulate the authors for their execution of this excellent manuscript.

Response: Thank you so much for your great efforts in the review of our ms and the excellent comments and recognition of our study.

Reviewer #3 (Remarks to the Author):

I maintain my position from previous review that the authors have conducted a very extensive study and identified interesting results that will be of notable interest. Various limitations were highlighted in the previous review that warranted clarification. The authors have made numerous additions and provided a very detailed response. My comments regarding these are detailed below.

The authors response to my comments as reviewer 3

- The human genetics evidence was not strengthened but the limitations of the study concerning the potential incidental nature of p.L700P are now very clearly acknowledged. I had not intended to indicate that statements such as “we have not performed rare variant association testing”, “the relatedness of the 3 patients was not assessed using available genotype data” and there was “no haplotype analysis” should be added to the text. If the authors have genotype data for the cohort that they report on and access to a bioinformatician with experience in population genetics then these tasks could likely be performed within the span of an afternoon. It is also now inconsistent that the authors add a line to their limitations section stating “the relatedness of the 3 patients was not assessed” while still referring to “three unrelated cases” throughout the manuscript. My assumption is that the authors indicate that there is “no known relationship between the 3 carriers”. However, the authors should confirm that these samples do not exhibit easily detected immediate familial relationships using KING (<https://www.kingrelatedness.com/>), plink (<https://www.cog-genomics.org/plink/1.9/ibd>) or an equivalent software package and the exome sequencing data. If it is not straight forward for the authors to conduct a rare variant association test then I would propose that they could also instead evaluate if the PCDHA9 locus is supported by homozygosity mapping (<https://www.homozygositymapper.org/>, again this can be done extremely quickly and could add more weight to the nomination of L700P). In my previous review I indicated that if the human genetics evidence could not be strengthened then the limitations should be stated more clearly, however quick checks such as these should be performed unless there is a specific reason that it is not possible.

Response: Thank you very much for the comments and very detailed professional advices. We agree that it is a very good point to make clear that the three patients carrying the homozygous L700P variant had no genetical relation among each other. These cases were identified from the panel sequencing data including 288 targeted

genes (see Results “Identification of a rare damaging homozygous PCDHA9 variant in Chinese sporadic ALS patients”, page 7, line 154-160). As we know, the immediate familial relationships cannot be evaluated using the panel sequencing data, since the number of variants identified in these data was too small. Unfortunately, the company to which we sent samples for sequencing many years ago did not retain the samples for further analysis suggested by the reviewers. However, we double-checked the birthplace and familial relationship of these patients based on at least three generations of information, and excluded the familial relatedness to each other.

- The observation of TDP-43 pathology is an extremely strong and interesting addition to the manuscript. The images look very compelling, but a formal quantification is lacking. Could the authors provide some kind of simple statistical comparison? Eg number of nuclei exhibiting TDP-43 depletion in mutant vs wildtype.

Response: Thanks for your advice. We provided the statistical analyze beside the images in figure 4f as shown below.

- I made some observations about inconsistencies in phenotypic severity concerning data in Fig2B and statements in the main text. The authors provide some explanation for this which concerns heterogeneity in the quality of housing conditions for animals assigned to different experiments. This is an important point as the severity of phenotypes is very relevant to the reader’s interpretation. The authors add one line to the methods but a more detailed statement **should be added to the main text** to clarify these discrepancies for the reader.

Response: Thank you very much for your advice. We put two lines in the main text accordingly in line 239-243 as: “Together, the above results demonstrate the progressive nature of the motor dysfunction phenotypes in Pcdha9 mutant mice. It is worth mentioning that Mut and Del mice used for the survival test under SPF conditions survived longer than those mice subjected to behavioral tests in clean-grade room. Thus, environmental factors may contribute to the behavioral phenotypes.”.

- I commented that methodological details describing the building of gene regulatory networks were not provided. The authors indicate that the approach is “very complicated” and “in

preparation for a ms". As a reviewer I can therefore offer no opinion as to whether the methodological approach for this section appears appropriate. However, as indicated in my previous review I also do not see that these sections of the manuscript are that critical to the overall story. They are interesting and relevant but do not yield insights that are substantiated and arguably receive too much attention. I think it is fine to include these analyses as tentative hypothesis generating analyses, so long as they are clearly indicated as such.

Response : We appreciate your detailed feedback on our manuscript. We have considered your comments and revised the sections related to gene regulatory networks. We also ensure that the exploratory nature of these analyses is clearly stated in accordance with your recommendations.

The method for the building of gene regulatory networks was developed by Dr. Qiang Tu's laboratory and is yet to be published. Please keep it confidential. The manuscript has been previously reviewed by Nature Methods and can be downloaded from <https://www.jianguoyun.com/p/DZuLrikQhuDnCBiq7e8DIAA>. As you can learn from the manuscript that the method is indeed complicated.

As requested, we put in a paragraph in the methods section to describe the method for constructing gene regulatory networks in much more detail (shown below). We have now clearly described the three-step process involved in the construction and analysis of the gene regulatory networks in line 990. We have further stated that our method for inferring cis-regulatory interactions, and hence defining a TF-gene GRN, is complex and therefore the insights gained from it are preliminary and hypothesis-generating.

'Construction of Gene Regulatory Networks. The construction and analysis of gene regulatory networks in our study involved a three-step process: (1) the examination of snATAC-seq and snRNA-seq datasets; (2) the integration of these datasets; and (3) the inference of cis-regulatory interactions to define a transcription factor-gene gene regulatory network (TF-gene GRN).

In terms of single-cell transcriptomes and epigenomes data integration, we employed the "addGeneIntegrationMatrix" function to incorporate the gene expression matrix of snRNA-seq data onto the "geneScoreMatrix" of snATAC-seq data in both WT and Mut mice.

The method utilized for the subsequent definition of TF-gene GRN, involving the inference of cis-regulatory interactions, is complex and a separate manuscript is currently being prepared in the laboratory of Dr. Qiang Tu. The insights gained from

GRNs of our study serve as preliminary findings and may provide potential directions for future hypothesis generation rather than as definitive conclusions.’

In addition, we have revised the relevant result sections, replacing definitive terms with phrases like "we generated potential gene regulatory networks", further emphasizing the exploratory nature of this aspect of our study in line 426.

We hope that these revisions address your concerns and improve the clarity of our manuscript. We appreciate your detailed review and valuable advices, which have undoubtedly helped enhance our work.

The authors response to issues raised by reviewer 2

- Clarified details concerning the breeding strategy should be added to the methods

Response: Details were added in method part at line 721-726 as advised. “The animals used, both knock-in and deletion mutants, were descendants of heterozygotes mating with heterozygotes. For the genotype of mice, we have sequenced the mouse DNA to verify the successful establishment of both Mut and Del mouse models using CRIPR/CAS9 (Figure S1c). They were kept separately to avoid mixed breeding. Genotype was inspected again for each descendant of those heterozygotes”.

- The rationale for the deletion line that was added on lines 215-217 is not entirely clear. Why is the fact that most of the variants identified in ExAC/gnomad are heterozygous relevant? Is this being interpreted as an indication of relevant selective constraint? If so gnomad generates predicted constraint metrics for genes in a more formal manner (pLI scores, available on their website), however the scores for PCDHA9 suggest reasonably high tolerance for loss of function of variants with multiple premature termination/frameshift variants occurring throughout the gene (including several observation of homozygous genotypes). In any case, as the authors suggest it seems relevant to contrast L729P (which may exhibit loss or gain of function) with a deletion variant for interpretation purposes. I feel this is the rationale that is required on lines 215-217 so that the reader is not given the impression that the deletion is being immediately treated as ALS relevant a priori.

Response: We are sorry for the confusion in explaining the rationale for using the deletion mouse line. The deletion line was not used as genetically ALS relevant animal model. Instead, it was used to see whether PCDHA9 L729P could be a loss of function mutation. We revised the sentence to “The deletion mouse line was used to inspect whether loss-of-function of PCDHA9 would result in phenotypes similar to *Pcdha9* L729P mutant.” In line 213-214.

- In my opinion a detailed longitudinal account of disease progression in the mouse models over time is not entirely clear. Reviewer 2 focused on this at length. Many comments were addressed, but interpretation remains complicated by the fact that mice kept in the clean grade

animal room exhibited poorer health than mice used for survival analyses. This disconnect between survival analyses and other phenotypic characterizations needs to be explained in the main text as otherwise the reader is guided to incorrect conclusions about the time course of events. However, I think the overall message that mice developed a robust late onset phenotype with evidence for progression and multiple key ALS associated hallmarks is already a clear compelling message that warrants reporting.

Response : Thanks a lot for your recognition of our efforts and conclusion. We explained the disconnect between survival analyses and other phenotypic characterizations in the main text as shown in line 240-243 as “It is worth mentioning that Mut and Del mice used for the survival test under SPF conditions survived longer than those mice subjected to behavioral tests in clean-grade room. Thus, environmental factors may contribute to the behavioral phenotypes.”

Reviewer #4 (Remarks to the Author):

The clarifications provided by the authors help in the understanding of the study itself and help to address some of the limitations raised by the reviewers. Overall, the homozygous and deletion mutants are helpful in demonstrating a mouse phenotype and the behavioral and phenotypic analyses are quite thorough for the symptomatic animals.

Major Concerns:

1. Given that PCDH gene clusters have been implicated in neurodevelopmental disorders (as the authors note in lines 112-116, additional pathological or electrophysiological analyses beyond the motor neuron counts performed at age 10 months would have provided additional insights into whether mutations or deletions in this gene were relevant in NMJ maintenance. Indeed, data in other models (including the SOD1 mouse model) suggest that there is a dying back phenomenon that predates motor neuron loss and phenotypic changes.

Response: This is a good point and thanks for your suggestion. We did not observe any abnormal behaviors in mutant mice before 2 months old, which is usually the time when most developmental disorders manifest abnormalities. That is why we systematically performed most of the motor behavior tests starting in 4 month-old mice. Based on the long-term behavioral tracking, we believe that 10-12 months should be the earliest ALS-like phenotypes onset time. We performed motor neuron (Chat positive neurons) counts at both age 10 and 12 months (Figure 4a and Figure S4a). Significantly reduced number was detected in 12 months old mutant mice but not in those 10 months old ones. These results implicate that there is no significant developmental defects in young mutant mice, at least, not significant enough to cause motor behavior defects.

However, consistent with the reviewer’s prediction of affected NMJ maintenance, our RNA-seq and ATAC-seq results from 10 months old mice showed that 38 genes encoding proteins located on the plasma membrane, synapse or cellular junctions

were disturbed between WT and Mut mice. These genes are likely to be involved in NMJ maintenance.

2. An important limitation to the study is the authors note in the response to reviewers that mice used in behavioral studies were kept in a clean grade room but those in the survival analysis were kept in the SPF room. This is an important finding that should be delineated in the “limitations of the study” section. This is noteworthy as other ALS models have also noted differences in phenotypes depending on the geographical and/or cleanliness of the animal rooms.

Response: Thank you for your advice. We agree with other reviewers that this very important and should be included in the main text in line 240-243 as: “It is worth mentioning that Mut and Del mice used for the survival test under SPF conditions survived longer than those mice subjected to behavioral tests in clean-grade room. Thus, environmental factors may contribute to the behavioral phenotypes.”.

3. The TDP immunostaining in figure 4 is unconvincing. The significant cytoplasmic staining (and aggregation) of TDP43 in the WT animals would be very atypical and it is not clear that it is particularly more abundant in the mutant animals. If the authors choose to speculate on TDP43 pathology/biology, there are a number of strategies that should be employed.

Response: We believe that the typical pathological feature of TDP-43 in our model is the nucleus clearance of TDP-43. The accumulation of TDP-43 in the cytoplasm maybe due to aging (12 month), but the cytoplasm accumulation in WT was weaker than that in Mut mice in most motor neurons. We provided the statistical analysis in Figure 4f in the revised ms.

4. The authors note in their response that mice that developed motor deficits all died. However, it is unclear why a large percentage of mice were unaffected either phenotypically or by survival. The responses of the authors are mostly restatements of the methods but don't offer insights into why such a large percentage of mice were unaffected and whether there are pathological correlates.

Response: This is a good point and it is common phenomenon in mouse disease models as mentioned above by the reviewer. There are variations among individual animals even in the wild type animals. As discussed above, we noticed that mice used for the inspection of survival rate under SPF conditions survived longer than those mice used in behavioral tests in clean-grade rooms. A large percentage of mice were unaffected either phenotypically or by survival in the survival analysis because they were kept in the SPF condition until 15 months. Even so, our result show statistically significant differences that is sufficient to indicate an association between gene mutations and disease. Due to the cost of mouse maintenance (1/3 of our lab budget is spent on animals), we have not inspected them at a much later time. We believe that those mutant mice would development phenotypes eventually. Meanwhile, most of

those analyzed for motor tests developed phenotypes earlier and more consistent because they were kept in clean-grade rooms. That is why we put in the main text as shown in line 240-243 as: “It is worth mentioning that Mut and Del mice used for the survival test under SPF conditions survived longer than those mice subjected to behavioral tests in clean-grade room. Thus, environmental factors may contribute to the behavioral phenotypes.” as suggested by the reviewer.

In fact, this may raise an intriguing question. Why most neurodegenerative disease happened at older age. In addition to genetic and environmental factors, aging plays an important role as well.

Minor concerns:

1. In lines 222-223, the authors may speculate on why paralysis started unilaterally in these mice.

Response: We would like to speculate that degeneration of motor neurons does not occur symmetrical on both sides of the extremities in both animals and patients. Another possibility is the compensatory effects of the left and right limbs are uneven.

2. The ChAT immunostaining in 4a and 4c look very different with a more diffuse staining in 4A (which I would expect) and a much more punctate staining in figure 4C (one wonders about autofluorescence).

Response: The reviewer has made the right judgement. The 4a is the result of ChAT immunostaining and 4c is the result of ChAT RNAscope in situ hybridization.

3. In lines 237-241, the authors speculate that differences in the swimming and rotarod test when compared to grip strength may have to do with aging. An alternative hypothesis may be that other non-motor pathways that correlate with balance may be affected and subsequently manifested in rotarod times.

Response: This is an excellent point. We fully agree that ‘other non-motor pathways that correlate with balance may be affected and subsequently manifested in rotarod times’. There is difference of sensitivity between behavioral tests. Both swimming and rota-rod tests are combination of long-term muscle strength and balance testing, while grip force is a test of short-term muscle strength. In the behavioral results of homozygous mutant mice, there are also differences in the onset time between different behavioral tests. Here, we just wanted to speculate that Pcdha9 heterozygous point mutation may have an effect on motor function, but it is present at a much later stage than homozygous mutant mice.

4. In lines 270-273 the authors argue that the finding of PCDHA9 by in situ hybridization is consistent with its important function in the survival of motor neurons. These data only suggest that it is expressed in motor neurons and does not implicate any importance in MN survival.

Response: We modified the sentence as “implicating the important function of Pcdha9

in motor neurons.” in line 272.

5. In lines 313-316, there is no quantification of WB to show that protein levels are “significantly” lower than that of WT.

Response: We agree with the point. We delete the “significantly” in the sentence, line 314-315, as “The protein levels of hPCDHA9 mutant was lower than that of WT, while the mRNA levels were similar (Fig. 6a-c).”.

REVIEWER COMMENTS

Reviewer #1 (Remarks to the Author):

In their study, entitled "Functional identification of PCDHA9 as a candidate causative gene for amyotrophic lateral sclerosis", Dr. Zhong and colleagues analyzed exome sequences in a cohort of Chinese ALS patients in search for novel disease-causing genes, uncovering a homozygous variant (p.L700P) in PCDHA9 in three unrelated patients that was absent from publicly available genomic databases. Pcdh α 9 mutant mice harbored either orthologous point mutation or deletion mutation and these mice developed progressive spinal motor loss, muscle atrophy, and structural/ functional abnormalities of the neuromuscular junction, leading to paralysis and early lethality. Of note, TDP-43 pathology was detected in spinal motor neurons of aged mutant mice. Mechanistically, the authors demonstrated that Pcdha9 mutation causes aberrant activation of FAK and PYK2 in aged spinal cord, and NKA- α 1 reduced expression in motor neurons. Single nucleus multi-omics analysis revealed disturbed signaling involved in cell adhesion, ion transport, synapse organization, and neuronal survival in aged mutant mice. The authors conclude that PCDHA9 is a new candidate ALS gene providing new insights into the pathogenesis of ALS. The study is well executed, and these results are of potential interest to the readership on Nature Communications. The authors have been responsive to the previous critic raised in the [REDACTED] submission; however, as this is a discovery of a new ALS gene with novel disease-causing variants, it is important to validate them in other patient cohorts via gene matchmaker or collaborative research. As the PCDHA gene cluster is large and complex gene cluster with marked redundancy it is difficult to determine functionality of any variants that is introduced or modeled. The gene contains multiple variable exons that are encoded in a complex way, with pre-mRNA generated by splicing making it difficult to endure specific gene/variant effects. Thus, the results presented herein remain preliminary and should be reported as such in the manuscript, indicating that further validation of PCDHA9 as a disease-causing gene in ALS is needed. Successful gene matchmaker hit or more advanced modeling studies showing phenotypic rescue or response to therapy targeting the MOA proposed would suffice for this.

Reviewer #3 (Remarks to the Author):

The authors have made every effort to respond to all concerns and discussion points raised during the review. I have no outstanding concerns and congratulate them on a fascinating study.

Kevin Kenna

(Signed as per transparency policy promoted by the journal)

Reviewer #5 (Remarks to the Author):

Major concerns:

1. The reviewer's comment here seemed to address two issues. One is whether there are developmental defects in the *Pcdha9* mutant and deletion mice that could lead to motor symptoms later in life, and two whether the dying back phenomenon observed in other ALS models including *SOD1* mice could also be observed prior to their earliest detection of motor neuron loss at 12 months. In other words, did the authors examine NMJ staining or EMG earlier than 12 months? The authors reason that since most developmental disorders manifest before 2 months of age and they did not observe abnormal behaviors within this window, they can rule out a deficiency in the developmental process in these mice. In the revised manuscript the authors do not present the data requested by the reviewer. Instead, they suggest that NMJ maintenance may be affected at 10 months based on their RNA-seq and ATAC-seq data sampled from that age. I agree that the reviewer's suggestion to examine the earlier ages (10 months or earlier) for deficits in NMJ maintenance can support the dying back hypothesis in parallel with the multi-omics data, and the outcome of this examination can support or refute this hypothesis. If the authors have any remaining samples at 10 months, they could quantify NMJ stainings. If there is no significant difference, they can refute a dying back model. If there is a significant difference, they can support it. If the authors cannot provide these data without having to breed more animals and wait 10 months, it would dampen the impact of this study in terms of offering a mechanistic model of motor neuron survival. Nevertheless, the observational phenotypes clearly support interest in *Pcdha9* as an ALS risk gene.

2. This concern was adequately addressed.

3. This concern was adequately addressed.

4. This concern was adequately addressed.

All minor concerns are addressed except for:

4 In my opinion, the sentence should rather be written: "We then used RNAscope® probes to conduct in situ hybridization assays and found strong *Pcdhα9* expression in motor neurons (marked by CHAT probe) in the WT lumbar spinal cord (Fig. 4c), suggesting specific function of *Pcdhα9* in motor neurons."

REVIEWER COMMENTS

Reviewer #1 (Remarks to the Author):

In their study, entitled "Functional identification of PCDHA9 as a candidate causative gene for amyotrophic lateral sclerosis", Dr. Zhong and colleagues analyzed exome sequences in a cohort of Chinese ALS patients in search for novel disease-causing genes, uncovering a homozygous variant (p.L700P) in PCDHA9 in three unrelated patients that was absent from publicly available genomic databases. Pcdha9 mutant mice harbored either orthologous point mutation or deletion mutation and these mice developed progressive spinal motor loss, muscle atrophy, and structural/ functional abnormalities of the neuromuscular junction, leading to paralysis and early lethality. Of note, TDP-43 pathology was detected in spinal motor neurons of aged mutant mice. Mechanistically, the authors demonstrated that Pcdha9 mutation causes aberrant activation of FAK and PYK2 in aged spinal cord, and NKA- α 1 reduced expression in motor neurons. Single nucleus multi-omics analysis revealed disturbed signaling involved in cell adhesion, ion transport, synapse organization, and neuronal survival in aged mutant mice. The authors conclude that PCDHA9 is a new candidate ALS gene providing new insights into the pathogenesis of ALS. The study is well executed, and these results are of potential interest to the readership on Nature Communications. The authors have been responsive to the previous critic raised in the [REDACTED] submission; however, as this is a discovery of a new ALS gene with novel disease-causing variants, it is important to validate them in other patient cohorts via gene matchmaker or collaborative research. As the PCDHA gene cluster is large and complex gene cluster with marked redundancy it is difficult to determine functionality of any variants that is introduced or modeled. The gene contains multiple variable exons that are encoded in a complex way, with pre-mRNA generated by splicing making it difficult to endure specific gene/variant effects. Thus, the results presented herein remain preliminary and should be reported as such in the manuscript, indicating that further validation of PCDHA9 as a disease-causing gene in ALS is needed. Successful gene matchmaker hit or more advanced modeling studies showing phenotypic rescue or response to therapy targeting the MOA proposed would suffice for this.

Response: Thank you very much for your great efforts in the review of our ms, and the excellent comments and recognition of our study.

We fully agree with the reviewer that 'it is important to validate them in other patient cohorts via gene matchmaker or collaborative research...'. As suggested, we identified two matching carriers of the homozygous PCDHA9 c.2099T>C (p.Leu700Pro, rs782621196) variant in the online GeneMatcher: <https://genematcher>.

org/reports/events? submissionID=58850&event Type=3. By correspondence with the providers of the information, we learned that both of the carriers were both Western children and the phenotype were ID/ASD/Rett-like syndrome (Match Identifier: 19S0538) and Benign infantile epilepsy, respectively. Although the phenotypes were different from that reported in our study, the matching results suggest that the homozygous L700P variants in PCDHA9 is pathogenic and may be associated with different phenotypes between children and adults. This statement has been added and highlighted in the 2 nd paragraph of Discussion.

We also recognized the point 'As the PCDHA gene cluster is large and complex gene cluster with marked redundancy it is difficult to determine functionality of any variants that is introduced or modeled...'. When we started this study, we have consulted with Professor Qian Wu from Dr. Maniatis' group who is one of the top experts on PCDHA family. We have ensured that the L700P mutation generated in mice is specific for the PCDHA9 isoform. The mutation is located in the middle of an exon specific for PCDHA9, not in other isoforms. The other part of the coding protein is consistent. Therefore, it is unlikely to affect other family members by splicing. Nevertheless, we cannot exclude the compensate effect of other genes in the PCDHA gene cluster for the homozygous PCDHA9 L700P mutation in mice.

We appreciate your advice 'Successful gene matchmaker hit or more advanced modeling studies showing phenotypic rescue or response to therapy targeting the MOA proposed would suffice for this'. As recommended, we are going to collaborate with Dr. Hui Yang in the Institute of Neuroscience, Chinese Academy of Sciences, to gene edit our PCDHA9 L700P mutant mice to confirm that the switching to WT PCDHA9 can rescue the ALS-like phenotype. In addition, we mentioned in the Limitations of the Study 'Thus, more cases should be inspected in other regions around the world, and further validation of PCDHA9 as a disease-causing gene in ALS is needed'.

Reviewer #3 (Remarks to the Author):

The authors have made every effort to respond to all concerns and discussion points raised during the review. I have no outstanding concerns and congratulate them on a fascinating study.

Kevin Kenna

(Signed as per transparency policy promoted by the journal)

Reviewer #5 (Remarks to the Author):

Major concerns:

1. The reviewer's comment here seemed to address two issues. One is whether there are developmental defects in the Pcdha9 mutant and deletion mice that could lead to motor symptoms later in life, and two whether the dying back phenomenon observed in other ALS models including SOD1 mice could also be observed prior to their earliest detection of motor neuron loss at 12 months. In other words, did the authors examine NMJ staining or EMG earlier than 12 months? The authors reason that since most developmental disorders manifest before 2 months of age and they did not observe abnormal behaviors within this window, they can rule out a deficiency in the developmental process in these mice. In the revised manuscript the authors do not present the data requested by the reviewer. Instead, they suggest that NMJ maintenance may be affected at 10 months based on their RNA-seq and ATAC-seq data sampled from that age. I agree that the reviewer's suggestion to examine the earlier ages (10 months or earlier) for deficits in NMJ maintenance can support the dying back hypothesis in parallel with the multi-omics data, and the outcome of this examination can support or refute this hypothesis. If the authors have any remaining samples at 10 months, they could quantify NMJ stainings. If there is no significant difference, they can refute a dying back model. If there is a significant difference, they can support it. If the authors cannot provide these data without having to breed more animals and wait 10 months, it would dampen the impact of this study in terms of offering a mechanistic model of motor neuron survival. Nevertheless, the observational phenotypes clearly support interest in Pcdha9 as an ALS risk gene.

Response: Thank you very much for your detailed review of our ms. We certainly agree with the point to test the dying back hypothesis. Because we do not have any remaining muscle samples left and our mice were too young at that time, we had not examined the NMJs at earlier age. Here, we inspected the NMJs in the 6-month-old wild type and homozygote Mut male mice and found that there is no significant difference between the two groups. The statistic results was added as supplementary figure 5e, and description was add in the main text at line 280-283. Although our results is not in support of the dying back hypothesis, we cannot refute it either as abnormal NMJ maintenance might happens at a later stage just before the death of motor neurons.

2. This concern was adequately addressed.

3. This concern was adequately addressed.

4. This concern was adequately addressed.

All minor concerns are addressed except for:

4 In my opinion, the sentence should rather be written: "We then used RNAscope® probes to conduct in situ hybridization assays and found strong Pcdha9 expression in motor neurons (marked by CHAT probe) in the WT lumbar spinal cord (Fig. 4c), suggesting specific function of Pcdha9 in motor neurons."

Response: Thank you for the advice. It is more accurate and we have modified the sentence as suggested. Line 272.

REVIEWERS' COMMENTS

Reviewer #1 (Remarks to the Author):

The authors have been responsive to the critique raised and have revised the manuscript accordingly and satisfactorily and outlined future functional studies. I have no further concerns.

Reviewer #5 (Remarks to the Author):

The authors have satisfactorily addressed all of my comments. The addition of the Gene Matcher results is interesting and helps contextualize the pathogenicity of the newly described mutation. The additional text suitably qualifies the impact of their data.